# Protein Language Model Embeddings Improve Generalization of Implicit Transfer Operators

Panagiotis Antoniadis [1]   Beatrice Pavesi [2]   Simon Olsson [* 2]   Ole Winther [* 1 3]

## Abstract

Molecular dynamics (MD) is a central computational tool in physics, chemistry, and biology, enabling quantitative prediction of experimental observables as expectations over high-dimensional molecular distributions such as Boltzmann distributions and transition densities. However, conventional MD is fundamentally limited by the high computational cost required to generate independent samples. Generative molecular dynamics (GenMD) has recently emerged as an alternative, learning surrogates of molecular distributions either from data or through interaction with energy models. While these methods enable efficient sampling, their transferability across molecular systems is often limited. In this work, we show that incorporating auxiliary sources of information can improve the data efficiency and generalization of transferable implicit transfer operators (TITO) for molecular dynamics. We find that coarse-grained TITO models are substantially more data-efficient than Boltzmann Emulators, and that incorporating protein language model (pLM) embeddings further improves out-of-distribution generalization. Our approach, PLaTITO, achieves state-of-the-art performance on equilibrium sampling benchmarks for out-of-distribution protein systems, including fast-folding proteins. We further study the impact of additional conditioning signals such as structural embeddings, temperature, and large-language-model-derived embeddings on model performance.

## 1. Introduction

Molecular dynamics (MD) simulations provide a computational bridge from microscopic physical laws to molecular phenomena, including those observed in biophysical experiments. Central to this task is sampling from high-dimensional molecular distributions, most notably Boltzmann equilibrium distributions and long-timescale transition densities. Classical MD relies on explicit numerical integration with femtosecond-scale time-steps, while relevant relaxation and mixing times span microseconds to seconds, resulting in prohibitive computational costs for all but the smallest systems. Recently, *generative molecular dynamics* (GenMD) methods have emerged (Olsson, 2026), which cast MD sampling as a generative modeling problem: neural networks are trained to produce independent samples from target molecular distributions, either from trajectory data or by learning from a potential energy model or force field. Existing GenMD approaches broadly fall into two classes: Boltzmann Generators/Emulators (Noé et al., 2019; Jing et al., 2022; Lewis et al., 2025) and Implicit Transfer Operators (Schreiner et al., 2023; Klein et al., 2023; Diez et al., 2025). Despite substantial gains in sampling efficiency, these methods typically require large collections of long MD trajectories, limiting data efficiency and generalization to unseen molecular systems.

We introduce coarse-grained transferable implicit transfer operators (TITO) for protein molecular dynamics that generalize to out-of-distribution protein systems when trained from scratch on diverse off-equilibrium MD trajectories across multiple temperatures (Mirarchi et al., 2024). To improve data efficiency, we incorporate representations from protein sequence and structure models, along with large language model–derived annotations. Among these variants, Protein-Language-aware TITO (PLaTITO, Figure 1) consistently achieves the best performance while retaining computational efficiency comparable to the base TITO model. Scaling model capacity further, we show that PLaTITO-19M achieves state-of-the-art results on equilibrium sampling benchmarks for out-of-distribution protein systems, including fast-folding proteins. Finally, we demonstrate

---

[*]Equal contribution, ordered alphabetically.  [1]Department of Biology, University of Copenhagen [2]Department of Computer Science and Engineering, Chalmers University of Technology and University of Gothenburg, SE-41296 Gothenburg, Sweden [3]DTU Compute, Technical University of Denmark. Correspondence to: Simon Olsson <simonols@chalmers.se>, Ole Winther <ole.winther@bio.ku.dk>.

*Proceedings of the 43rd International Conference on Machine Learning*, Seoul, South Korea. PMLR 306, 2026. Copyright 2026 by the author(s).

---

[1]Code is available at: https://github.com/PanosAntoniadis/platito

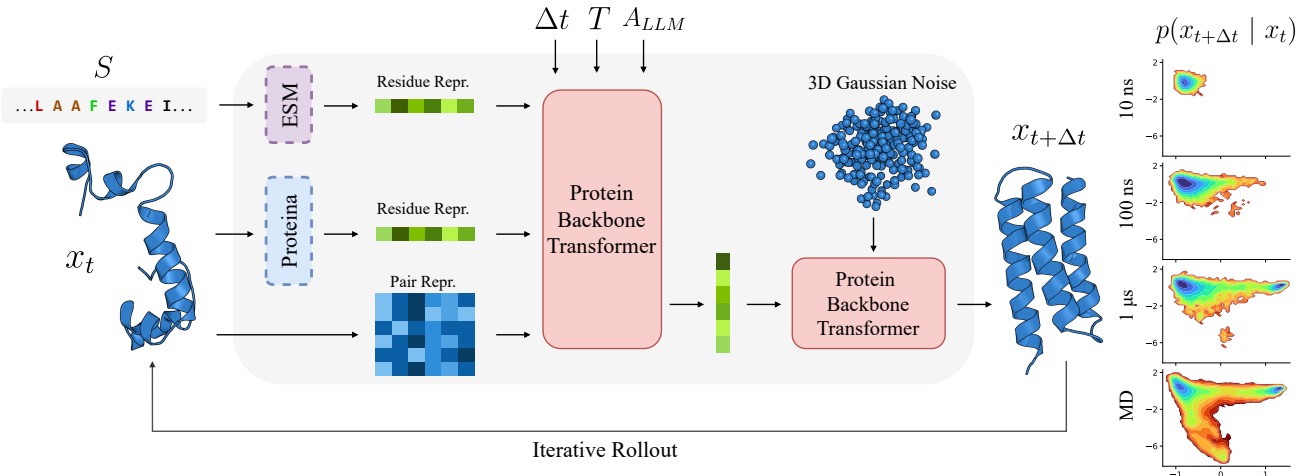

*Figure 1.* **PLaTITO generalizes to unseen protein systems while improving data efficiency.** Given the molecular state of a protein system at physical time $t$, defined by backbone coordinates $x_t$, amino-acid sequence $S$ and temperature $T$, our proposed TITO models approximate the long-time transition density $p(x_{t+\Delta t} \mid x_t, S, T, \Delta t)$ for a given time step $\Delta t$. To improve data efficiency, auxiliary representations are incorporated during training including pretrained sequence embeddings from ESM, pretrained structure embeddings from Proteina and LLM-derived annotations $A_{LLM}$. Iterative sampling of the learned transition model enables sampling of protein conformational dynamics at increasing timescales approaching the equilibrium distribution of the MD.

that our temperature-dependent TITO model recovers non-Arrhenius protein folding and unfolding rates.

Our main contributions are:

1. We develop coarse-grained transferable implicit transfer operators (TITO) for protein molecular dynamics that **generalize to out-of-distribution protein systems** without system-specific fine-tuning.

2. We investigate the impact of conditioning on auxiliary sources of information to boost data efficiency. We introduce a Protein-Language-aware TITO (PLaTITO) that achieves **state-of-the-art performance on equilibrium sampling benchmarks** while being trained with substantially less data and computational resources.

3. We show that the temperature-dependent kinetics learned by PLaTITO are non-Arrhenius, **consistent with complex rugged folding free energy landscapes**.

## 2. Background and Preliminaries

Section 2.1 introduces Molecular Dynamics, section 2.2 presents flow matching as a density estimation approach and Section 2.3 introduces ITO for learning long-timescale transition densities, addressing a key problem outlined in Section 2.1.

### 2.1. Molecular dynamics and observables

Molecular dynamics (MD) simulations typically evolve molecular configurations $x \in \Omega$ by numerically integrating

the Langevin equation (Langevin et al., 1908) under a potential energy $U(x)$. This generates a discrete-time Markov process with transition density $p(x_{t+\tau} \mid x_t)$, where the step size $\tau$ must remain small for numerical stability. Under certain conditions, this process admits the Boltzmann distribution, $\mu(x) \propto \exp(-\beta U(x))$, as its invariant measure. Physical observables are then computed as expectations $\mathbb{E}_\mu[f(x)]$ for stationary properties (e.g., binding affinities) or time-correlations for dynamical properties (e.g., folding/unfolding rates). However, because $\tau$ is restricted to the femtosecond scale, standard MD requires prohibitive simulation lengths to bridge the gap to biologically relevant timescales, leading to biased observable estimates, necessitating the development of accelerated surrogate models.

### 2.2. Flow Matching

Flow-matching (Albergo & Vanden-Eijnden, 2023; Liu et al., 2023; Lipman et al., 2023) is a generative modeling framework that approximates a target distribution by learning a continuous density path $p_s(z)$, $s \in [0, 1]$ that transforms a simple base distribution $p_0(z)$ to an approximation of the data distribution $p_1(z) \approx p_{\text{data}}(z)$. Formally, the density path is induced by a flow map $\phi_s$ such that $p_s = [\phi_s]_\# p_0$ where $\#$ denotes the push-forward operator. The flow map is defined as the solution of an ordinary differential equation (ODE)

$$\frac{d}{ds} \phi_s(z) = u_s\big(\phi_s(z)\big), \quad \phi_0(z) = z \qquad (1)$$

where $u_s$ denotes the time-dependent velocity field parameterized by the flow-matching time $s$.

To learn an approximation of $u_s$, we adopt Conditional

Flow Matching (CFM) that constructs conditional probability paths $p_s(z \mid z_1)$ conditioned on data samples $z_1 \sim p_1$ where the respective velocity field $u_s(z \mid z_1)$ is analytically tractable. The model is trained by regressing a neural vector field $v_s^\theta$ against the conditional velocity field:

$$\mathcal{L}(\theta) = \mathbb{E}_{s, z_1 \sim p_1, z \sim p_s(\cdot \mid z_1)} \left[ \left\| v_s^\theta(z) - u_s(z \mid z_1) \right\|^2 \right] \quad (2)$$

We adopt the linear conditional paths, described in the rectified flow formulation (Liu et al., 2023), $z_s = s z_1 + (1-s) z_0$, that generate a constant conditional velocity field $u_s(z \mid z_1) = z_1 - z_0$.

### 2.3. Implicit Transfer Operator Learning

Implicit Transfer Operator (ITO) learning (Schreiner et al., 2023) provides a framework for modeling long-timestep molecular dynamics by learning surrogate models of transition densities $p(x_{N\tau} \mid x_0)$ from MD data, where $N$ denotes a discrete time step. In practice, we train a generative model $x_{t+N\tau} \sim p_\theta(x_{t+N\tau} \mid x_t, N)$ using tuples $(x_{t_i}, x_{t_i + N_i \tau}, N_i) \in (\mathbb{R}^{3L}, \mathbb{R}^{3L}, \mathbb{N})$ randomly sampled from molecular dynamics trajectories of $L$ particles in 3D, to approximate the transition probability $p(x_{N\tau} \mid x_0)$ at multiple time-steps.

## 3. Related Work

**Boltzmann Generators and Emulators** Boltzmann Generators (BG) are surrogates of the Boltzmann distribution, $\mu(x)$, learned from data, potential energy model or a combination of the two, and allow for exact reweighting to target Boltzmann density (Noé et al., 2019). There are variants that transfer across chemical space (Klein & Noé, 2024; Tan et al., 2025; Akhound-Sadegh et al., 2025) and thermodynamic state (Moqvist et al., 2025; Schebek et al., 2024). Boltzmann Emulators (BE) (Jing et al., 2022; Lewis et al., 2025; Diez et al., 2024; Daigavane et al., 2026) sacrifice details and theoretical guarantees and instead focus on scaling and external validation. Other approaches include conservative diffusion-based models that allow sampling (Wang et al., 2026) or simulation (Arts et al., 2023; Plainer et al., 2025). Other related methods were surveyed recently (Janson & Feig, 2025; Bonneau et al., 2026; Olsson, 2026).

**Transfer Operator surrogates** Approximating molecular transition densities using transfer operator-based approaches has been widely studied including Markov state models (MSMs) (Prinz et al., 2011), dynamic graphical models (Olsson & Noé, 2019; Hempel et al., 2022), observable operator models (Wu et al., 2015), and VAMPnets (Mardt et al., 2018). MSMs approximate the transfer operator through a discrete time and space representation while VAMPnets use a neural network to learn a membership function to a discrete number of states through the variational approach

for Markov processes (VAMP) (Wu & Noé, 2019). More recently, generative models have been proposed to approximate the transfer operator by learning from MD trajectories. Timewarp (Klein et al., 2023) is a flow-based model that provides unbiased equilibrium samples through a Metropolis-Hastings correction with limited transferability on small peptides. ITO (Schreiner et al., 2023) introduced the use of conditional diffusion models to build multiple time-scale MD surrogate models and its successor TITO (Diez et al., 2026) demonstrated robust transferability across unseen peptides and small molecules, on ultra-slow time-scales, even when trained on short off-equilibrium MD trajectories. Diez et al. (2025) integrated Boltzmann priors into the ITO framework to enforce asymptotically unbiased equilibrium statistics. Fu et al. (2023) introduced a multi-scale graph neural network to predict deterministic long-time displacements of coarse grained polymers, with a diffusion based correction, similar to later flow-based models of atomic transport (Nam et al., 2025). Costa et al. (2025a) used two-sided stochastic interpolants (Albergo & Vanden-Eijnden, 2023) to directly model $p(x_{t+N\tau} \mid x_t)$ without a latent Gaussian and a fixed $N$. Later, the authors extended this work in DeepJump (Costa et al., 2025b) training on the larger mdCATH dataset (Mirarchi et al., 2024) illustrating limited transferability to the fast-folders. Other approaches include those which ignore the Markovian structure when modeling MD trajectories (Vlachas et al., 2021; Jing et al., 2024; Murtada et al., 2025). In this work, we introduce a coarse-grained, non-equivariant model that scales the ITO learning framework in both model capacity and dataset size and incorporates additional conditioning signal from pre-trained models to achieve state-of-the-art performance on equilibrium sampling benchmarks.

## 4. Methods

### 4.1. Flow Matching for ITO learning with multiple conditioning

Building upon the ITO learning framework, our goal is to train a model to approximate the long-timescale transition density $p(x_{t+\Delta t} \mid x_t, \Delta t, S, T)$ where $\Delta t$ denotes the time step, $S$ the amino-acid sequence of a protein and $T$ the simulation temperature. The configurations $x_{t+\Delta t}, x_t$ correspond to protein coordinates drawn from reference trajectories $\mathcal{X} = \{\ldots, x_t, x_{t+\tau}, \ldots\} = \{x_{k\tau}\}_{k=0}^M$ generated by explicit MD simulation using a femtosecond-scale integration step $\tau$. To enable scaling, we adopt a coarse-grained representation of the proteins using only the $C_\alpha$ backbone coordinates such that $x_t \in \mathbb{R}^{3L}$ where $L$ is the total number of residues.

Following Schreiner et al. (2023) we sample pairs of backbone coordinates $x_t$ and $x_{t+\Delta t}$, randomly from the same trajectory spaced in time $\Delta t \gg \tau$. We introduce a la-

tent flow variable $z_s$, $s \in [0, 1]$, that interpolates between Gaussian noise and the future conformation, with $z_1 = x_{t+\Delta t} \sim \mathcal{X}$ and $z_0 \sim \mathcal{N}(0, I)$. We define a linear interpolant $z_s = s\, z_1 + (1 - s)\, z_0$ and we learn the target velocity field $v_s = z_1 - z_0 = x_{t+\Delta t} - z_0$ by minimizing the conditional flow-matching objective:

$$\mathcal{L}(\theta) = \mathbb{E}_{x_t, x_{t+\Delta t} \sim \mathcal{X},\ s \sim \mathcal{U}(0,1),\ z_0 \sim \mathcal{N}(0,I)}$$
$$\left\| v^\theta(z_s\ ;s, x_t, \Delta t, S, T) - (x_{t+\Delta t} - z_0) \right\|^2 \quad (3)$$

where $S$ denotes the amino-acid sequence of the protein and $T$ denotes the simulation temperature in Kelvin. During training, we sample proteins across multiple temperatures and time-steps to expose the model to diverse conditions, thereby improving generalization to unseen systems.

## 4.2. Architecture

Our architecture follows a two-stage design that separates conditioning on the system's state at time $t$ from the prediction of the velocity field (Figure 1). In the first stage, we compute a per-residue conditioning representation

$$c = f_c(x_t,\ \Delta t,\ S,\ T) \quad (4)$$

that takes as input the molecular state $x_t \in \mathbb{R}^{3L}$, the amino-acid sequence $S$, the simulation temperature $T$ and the time step $\Delta t$. In the second stage, a separate network $f_v$ predicts the velocity field at flow-matching time $s$ as $v = f_v(z_s, s, c)$. Both $f_c$ and $f_v$ are non-equivariant transformer architectures based on Proteina (Geffner et al., 2025). They construct residue and pair representations from the input 3D backbone coordinates, residue indices and sequence separation between residues using $x_t$ for the conditioning network $f_c$ and $z_s$ for the velocity network $f_v$. Learnable embeddings for the conditioning variables $\Delta t$, $S$, and $T$ (in $f_c$) and for $s$ and $c$ (in $f_v$) are concatenated to the residue representations, as described in Appendix A. Then, the residue representations are processed by a stack of conditioned and biased multi-head self-attention layers (Vaswani et al., 2017), where pair representations are used as attention biases. After the final transformer layer, the residue representations are decoded by $f_v$ into the velocity field prediction $v \in \mathbb{R}^{3L}$, while the conditioning representation $c \in \mathbb{R}^{L \times \dim}$ is obtained from the final residue representations produced by the conditioning network $f_c$. More details on the architecture and on the training and sampling procedures are available in Appendices A and B. Models trained with this architecture are referred to as **TITO** in Table 1.

## 4.3. Incorporating auxiliary representations

To evaluate the impact of other sources of conditioning signal on transferability in ITO learning, we augment the model conditions with three additional sources of information:

**I. Sequence embeddings** Protein Language Models (pLMs), trained on billions of unlabeled protein sequences, have been shown to implicitly capture a wide range of protein features in their latent representations including evolutionary, structural and functional information (Lin et al., 2023; Hayes et al., 2025). We leverage learned pLM representations as a computationally efficient source of prior knowledge for our TITO models. Given an input sequence $S \in \mathcal{A}^L$ of length $L$ where $\mathcal{A}$ denotes the amino-acid alphabet, we extract residue-level embeddings using a pretrained pLM as $e_{\text{seq}} = \phi_{\text{pLM}}(S) \in \mathbb{R}^{L \times d_{\text{pLM}}}$. These embeddings are provided as additional inputs to the conditioning network and can be precomputed offline, introducing no additional computational overhead during model training. Models trained with this architecture are referred to as **PLaTITO** or **PLaTITO-19M** in Table 1.

**II. Structure embeddings** To incorporate prior structural information into our MD surrogate models, we leverage structure-aware representations extracted from models trained on protein structures. Recent generative models have demonstrated strong performance in protein backbone generation (Geffner et al., 2025; Lin et al., 2024) by scaling training to synthetic structures from AFDB (Varadi et al., 2022). We suggest that the learned representations of these models contain local and global geometric features that can prove useful for modeling dynamics. Given a backbone configuration $x_t \in \mathbb{R}^{3L}$, we extract residue-level embeddings using a pretrained structure-aware model as $e_{\text{struct}} = \phi_{\text{PSM}}(x_t) \in \mathbb{R}^{L \times d_{\text{PSM}}}$. Unlike sequence embeddings that can be precomputed offline, structure embeddings must be computed online for each generated backbone configuration, making the efficiency of $\phi_{\text{PSM}}$ critical. Models trained with this architecture are referred to as **PLaTITO+Struct** in Table 1.

**III. LLM-derived annotations** In MD datasets, protein domains are often extracted from their native structural environment and simulated in isolation for computational efficiency (Mirarchi et al., 2024). As a result, some simulations can exhibit unphysical behavior due to missing binding partners, cofactors or contextual constraints. Since manually annotating such cases does not scale to large datasets, we propose using large language models (LLMs) as a heuristic tool to assess whether a given simulation is likely to reflect the native behavior of a protein system despite the absence of its full structural context. For each system, we extract a curated set of metadata capturing structural and biological context, including domain length, macromolecular composition, subcellular localization and known functional annotations. This metadata is used to prompt an LLM to assess the suitability of simulating the protein domain in isolation. For each system, we obtain a set of annotation features $A_{LLM} = \{S_{LLM}, C_{LLM}\}$ that consists of a bi-

*Table 1.* **PLaTITO-19M achieves state-of-the-art performance on equilibrium sampling while requiring substantially less MD training data and computational resources.** Equilibrium sampling validation metrics comparing equilibrium distributions generated by our TITO models to MD reference distributions and to the BioEmu model for fast-folding proteins, together with number of trainable parameters, training data size and computational cost.

| Model | MAE ($\downarrow$) | RMSE ($\downarrow$) | Coverage ($\uparrow$) | GPU hours[1] | MD data | Parameters |
|---|---|---|---|---|---|---|
| TITO | $1.068\pm0.272$ | $1.382\pm0.302$ | $0.590\pm0.111$ | 1100 | 56 ms | $3M$ |
| TITO+Struct | $1.004\pm0.290$ | $1.310\pm0.350$ | $0.560\pm0.134$ | 1100 | 56 ms | $3M$ |
| PLaTITO | $0.949\pm0.269$ | $1.228\pm0.328$ | $0.651\pm0.151$ | 1100 | 56 ms | $3M$ |
| PLaTITO+Struct | $0.938\pm0.321$ | $1.213\pm0.348$ | $0.655\pm0.158$ | 1100 | 56 ms | $3M$ |
| PLaTITO+Struct+LLM | $1.066\pm0.270$ | $1.346\pm0.292$ | $0.570\pm0.087$ | 1100 | 56 ms | $3M$ |
| PLaTITO-19M | $\mathbf{0.824\pm0.170}$ | $\mathbf{1.099\pm0.212}$ | $\mathbf{0.666\pm0.136}$ | 1100 | 56 ms | $19M$ |
| Emu | $1.305\pm0.378$ | $1.639\pm0.406$ | $0.529\pm0.112$ | 1100 | 56 ms | $3M$ |
| BioEmu[2] | $1.110\pm0.292$ | $1.389\pm0.346$ | $0.594\pm0.175$ | 9216 | 216 ms | $31M$ |

[1] GPU hours for training measured on a single NVIDIA A100 80GB GPU.
[2] BioEmu was additionally trained on 131k AFDB structures and 502k experimental $\Delta G$ measurements.

nary suitability label $S_{LLM}$ indicating whether the protein is expected to remain stable under the given MD conditions and an associated confidence score $C_{LLM}$ (More details in Appendix C). Models trained with this architecture are referred to as **PLaTITO+Struct+LLM** in Table 1.

## 4.4. Dataset

We train our TITO models on the mdCATH dataset (Mirarchi et al., 2024) that consists of diverse off-equilibrium MD trajectories across multiple temperatures, enabling the training of temperature-dependent models with a potential to generalize. We restrict the training set to protein domains of at most 200 residues, resulting in 4,482 domains. To ensure a strict train-test split, we remove any protein with at least 40% sequence similarity to a test protein over an alignment of at least 20 residues following the procedure of (Lewis et al., 2025). Our filtered training set includes 4,471 domains, corresponding to approximately 56 ms of aggregate MD simulation time.

## 5. Results

### 5.1. Models

To evaluate the impact of incorporating external representations on generalization and data efficiency, we train five model variants using the same dataset split and training compute. All models learn to approximate the long-time transition distribution $p(x_{t+\Delta t} \mid \cdot)$. We consider the following variants:

1. **TITO**. A baseline 3M-parameter model trained without any external representation conditioned on the current backbone coordinates, sequence, temperature, and time step:
$$p(x_{t+\Delta t} \mid x_t, \Delta t, S, T) \qquad (5)$$

2. **TITO+Struct**. A Protein Structure-aware variant of TITO that conditions on structure embeddings from the

pretrained Proteina 60M model (Geffner et al., 2025):
$$p(x_{t+\Delta t} \mid x_t, \Delta t, e_{struct}, S, T) \qquad (6)$$

3. **PLaTITO**. A Protein Language–aware variant of TITO that conditions on sequence embeddings extracted from the pretrained ESM Cambrian 300M pLM (ESM Team, 2024):
$$p(x_{t+\Delta t} \mid x_t, \Delta t, e_{seq}, T) \qquad (7)$$

4. **PLaTITO+Struct**. An extension of PLaTITO that further conditions on structure embeddings from the pretrained Proteina 60M model (Geffner et al., 2025):
$$p(x_{t+\Delta t} \mid x_t, \Delta t, e_{struct}, e_{seq}, T) \qquad (8)$$

5. **PLaTITO+Struct+LLM**. A further extension of PLaTITO+Struct that conditions on LLM-derived annotations, including suitability and confidence embeddings obtained by prompting the DeepSeek Reasoner LLM (Guo et al., 2025):
$$p(x_{t+\Delta t} \mid x_t, \Delta t, e_{struct}, e_{seq}, T, A_{LLM}) \qquad (9)$$

### 5.2. Equilibrium Sampling

We first evaluate the ability of our models to reproduce the equilibrium distribution of the fast-folding proteins (Lindorff-Larsen et al., 2011). The dataset consists of 12 systems ranging from 10 to 80 residues, simulated with the CHARMM22* force field (Piana et al., 2011) in explicit solvent, with atomic configurations saved every 200 ps. The data is proprietary but available upon request for research purposes.

For each system, we estimate a Time-lagged Independent Component Analysis (TICA) model (Pérez-Hernández et al., 2013) on pairwise $C_\alpha$-$C_\alpha$ distances computed from the reference MD trajectories and a lag time of 10 ns. The reference trajectories are then projected into the four slowest

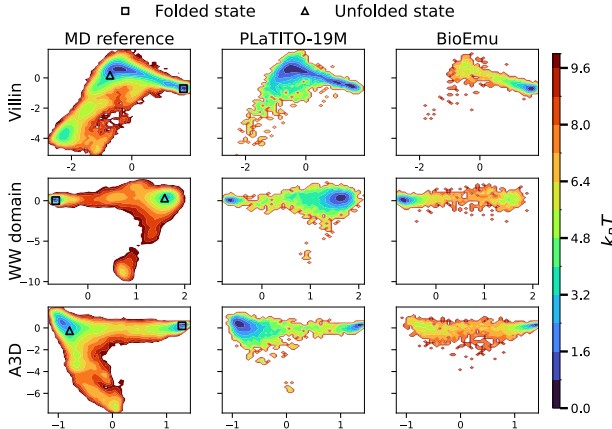

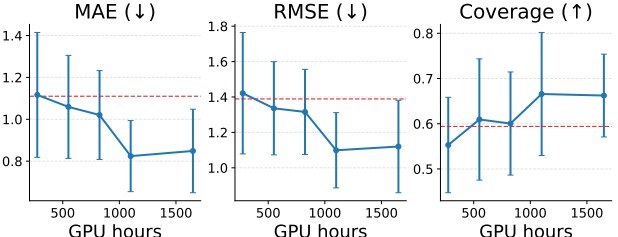

*Figure 3.* **Scalability of PLaTITO-19M with training compute.** Equilibrium sampling metrics improve as training compute increases, indicating effective scaling behavior. The red dashed line corresponds to the performance of BioEmu. Notably, PLaTITO-19M converges within approximately 1,100 GPU hours, that is substantially less than the training cost required by BioEmu (9,216 GPU hours) highlighting the computational efficiency of TITO models compared to Boltzmann Emulators.

*Figure 2.* **Test-time predictions of free energy landscapes of three fast-folders.** Free energy surfaces projected into the two slowest TICA components of Villin, WW domain and A3D. PLaTITO-19M (middle) accurately reproduces the MD reference distributions (left) and exceeds the performance of BioEmu (right). Squares (□) and triangles (△) denote folded and unfolded states, respectively, with PLaTITO-19M trajectories initialized from the unfolded state. Results for all fast-folding proteins are shown in Appendix D.2.

TICA components, yielding a low-dimensional representation of the equilibrium distribution. To estimate free-energy landscapes, the projected trajectories are discretized into bins and a normalized histogram $p_i$ is computed. The corresponding free energy for each bin $i$ is then computed as

$$G_i = -k_B T \ln p_i \tag{10}$$

where $p_i$ is the normalized histogram count for bin $i$, $k_B$ is the Boltzmann constant and $T$ is the simulation temperature. We initialize 1,000 trajectories from an unfolded state and perform 1,000 iterative roll-outs with a physical time step $\Delta t = 1\,\text{ns}$, resulting in 1 µs trajectories, since the mean transition-path time of folding $\tau_p$ is on the order of 1 µs for all the systems (Lindorff-Larsen et al., 2011). To approximate the stationary distribution, we keep only the final 10 ns of each trajectory resulting in 10,000 generated samples per system. These samples are projected in the reference TICA coordinates and converted to free-energy estimates following the same procedure as for the MD reference. Sampling hyperparameters are summarized in Table 3 and the resulting free-energy landscapes are shown in Appendix D.1.

In Table 1, we report the mean absolute error (MAE) and root mean squared error (RMSE) between the free-energy surfaces predicted by the models and the MD reference (detailed description of the metrics in Appendix B.5). In addition, we report Coverage, defined as the fraction of bins in reference MD that are sampled by the model (Diez et al., 2026). We observe that PLaTITO leads to a substan-

tial improvement over the base TITO model, reducing both MAE and RMSE while increasing Coverage at no additional test-time cost. This result aligns with previous work which suggests that pretrained pLM representations encode useful thermodynamic information (Meier et al., 2021; Frazer et al., 2021; Frellsen et al., 2026) which in turn can be leveraged to improve data efficiency in ITO learning. Incorporating structure embeddings (PLaTITO+Struct) provides a modest but consistent improvement across all metrics, indicating that structure and language embeddings provide complementary information. In contrast, conditioning on LLM-derived annotations (PLaTITO+Struct+LLM) decreases performance suggesting that either the prompting strategy or the available metadata were insufficient to provide useful conditioning signals to the model.

To evaluate the impact of model capacity, we implement **PLaTITO-19M** that is a 19M-parameter variant of the PLaTITO model leveraging pLM embeddings from the larger ESM Cambrian 6B pLM. PLaTITO-19M achieves further improvements in equilibrium sampling under the same training compute budget, highlighting the benefits of scaling both model capacity and pretrained sequence representations (Figure 3). We compare PLaTITO-19M to BioEmu (Lewis et al., 2025), a recently proposed Boltzmann emulator that achieved state-of-the-art performance in reproducing equilibrium distributions of MD trajectories. *PLaTITO-19M outperforms BioEmu across all equilibrium sampling evaluation metrics* despite being trained with substantially less data and a lower compute budget. We provide additional comparisons with ConfDiff (Wang et al., 2024) and Str2Str (Lu et al., 2024) in Appendix D.5, where PLaTITO-19M maintains the best performance across all models. Finally, in Appendices D.3 and D.4 we investigate various pLMs and model sizes, illustrating that pLM embeddings consistently improve generalization regardless of the model family, and that scaling pLM representations and model capacity provide complementary benefits.

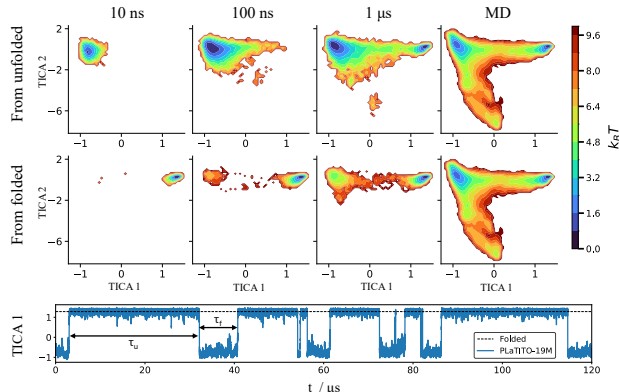

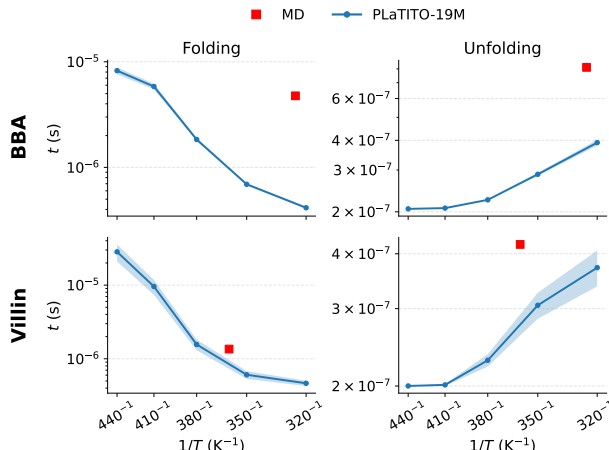

*Figure 4.* Top: Free-energy surfaces projected into the two slowest TICA components of A3D estimated by PLaTITO-19M from 1,000 independent trajectories initialized either from unfolded (top row) or folded (middle) conformations. Distributions are shown at increasing rollout times (left to right) and compared to the MD reference distribution (rightmost column). Below: Time-trace of a long 120 μs trajectory by PLaTITO-19M projected in the slowest TICA component, illustrating repeated folding and unfolding events. Results for all fast-folding proteins are shown in Appendix D.6.

Notably, even our smaller TITO variants achieve performance comparable to BioEmu, highlighting that TITO models are substantially more data-efficient than Boltzmann Emulators while achieving competitive equilibrium sampling performance. To further demonstrate this, we trained Emu, a model architecturally identical to TITO but trained as a Boltzmann Emulator to sample directly from $p(x|S, T)$. TITO outperforms Emu across all metrics despite using exactly the same architecture, data and compute budget indicating that exploiting the auto-correlation structure of MD data lowers the data and compute requirements of training MD surrogates.

In Figure 2, the free energy landscapes for a representative subset of three fast-folding proteins (Villin, WW domain, and A3D) are shown, where PLaTITO-19M accurately reproduces both the folded and unfolded parts of the reference MD free-energy landscapes and consistently outperforms BioEmu. We report a comprehensive evaluation across all fast-folding proteins in Appendix D.2 and we provide additional details on BioEmu sampling in Appendix B.2.

### 5.3. Long time-scale dynamics

Beyond equilibrium sampling, ITO models can generate long time-scale dynamics by sampling from the learned transition densities $p(x_{\Delta t} \mid x_0)$. We therefore evaluate whether PLaTITO-19M can reproduce folding–unfolding transition kinetics, rather than merely emulating equilibrium distributions. To this end, we generate 1,000 independent trajectories initialized either from unfolded or folded conformations

*Figure 5.* **PLaTITO-19M recovers non-Arrhenius folding and unfolding rates**. Folding (left) and unfolding (right) timescales predicted by PLaTITO-19M are shown as a function of inverse temperature for BBA (top) and Villin (bottom). The predicted rates exhibit clear deviations from simple Arrhenius behavior indicating that the learned temperature-conditioned dynamics capture physically meaningful kinetic trends. Reference rates estimated from MD simulations are shown as red squares.

and observe $p(x_{\Delta t} \mid x_0)$ at increasing rollout times (Figure 4, top). In both cases, PLaTITO-19M progressively explores intermediate states converging to a stationary distribution, approximating the reference MD distribution better than any baseline. We further generate a long 120 μs trajectory for A3D starting from an unfolded state using iterative rollouts of PLaTITO-19M. The generated time-trace exhibits repeated folding and unfolding events, indicating that the model captures the slow dynamics of the system (Figure 4, bottom). Results for all the fast-folders are available in Appendix D.6. The estimated mean first passage times (MFPT) of folding and unfolding ($\langle\tau_f\rangle^{PLaTITO} = 5.5 \pm 0.9\,\mu s$ and $\langle\tau_u\rangle^{PLaTITO} = 14.8 \pm 1.2\,\mu s$) are faster than the corresponding MD estimates ($\langle\tau_f\rangle^{MD} = 27 \pm 8\,\mu s$ and $\langle\tau_u\rangle^{MD} = 31 \pm 9\,\mu s$) (Lindorff-Larsen et al., 2011) which aligns with our expectations as a variational principle suggests that imperfect approximations of MD systematically underestimate time-scales (Nüske et al., 2014).

### 5.4. Temperature-dependent rates

Since our proposed TITO models are explicitly conditioned on simulation temperature $T$ during training, we evaluate whether the learned operator predicts physically plausible temperature-dependent rates of folding and unfolding. Since protein folding is characterized by complex high-dimensional free energy landscapes populated with many transient intermediate structures (Scalley & Baker, 1997; Kragelund et al., 1999), we expect deviation from the idealized exponential temperature dependence in the activation energy $E_a$ described by Arrhenius (1889),

$k(T) = A \exp\left(-E_a/k_B T\right)$, where $A$ is a pre-exponential factor and $k_B$ is Boltzmann's constant.

For five different temperatures ranging from 320 K to 440 K, we generate 1,000 trajectories of total duration 1 µs initialized from an unfolded state and estimate folding and unfolding rates using mean first-passage times (MFPTs) following Schreiner et al. (2023)

In Figure 5, we report the predicted folding and unfolding timescales, $t = 1/k$, as functions of inverse temperature $1/T$. It is clear that PLaTITO-19M exhibits a non-Arrhenius temperature dependence, consistent with prior studies of protein folding kinetics (Alexander et al., 1992; Tan et al., 1996; Schindler & Schmid, 1996; Wang et al., 2003). Further, we find that reference timescales from MD at selected temperatures (red squares, Figure 5) are consistently larger than predictions from PLaTITO-19M, aligning with the variational principle of conformational dynamics (Nüske et al., 2014), and suggesting that further scaling of data or model size may improve predictions further.

### 5.5. Formation of cryptic binding pockets

We evaluate the ability of PLaTITO-19M to explore conformations connected to cryptic binding pockets: lowly populated conformational states of proteins which might be targeted by therapeutics (Raich et al., 2021; Zhang & Bowman, 2026). We evaluate four experimentally characterized cases curated by Lewis et al. (2025). For each system, we generate 100 trajectories initialized either from the *apo* or from the *holo* state and perform 1,000-step roll-outs with $\Delta t = 1 \, \mathrm{ns}$ and simulation temperature $T = 350 \, \mathrm{K}$. Following Lewis et al. (2025), we compute local $C_\alpha$ RMSDs, using only binding pocket residues to the reference *apo* and *holo* structures.

In Figure 6 we show results for two representative systems (other systems are reported in Appendix D.7). In all cases, we sample broad ensembles with the majority of the probability density being assigned to conformations which are distinct from both *apo* and *holo* states. When initialized from the *holo* state, PLaTITO-19M generates samples similar to the *apo* state, demonstrating improved coverage compared to the baseline. While BioEmu fails to sample the *apo* state in three out of four benchmark cases, PLaTITO-19M improves upon this by recovering the *apo* state in one of these difficult instances, failing in only two of the three baseline failure cases (systems Q29495 and Q58L87, Appendix D.7). We note that in these remaining failure modes, PLaTITO-19M does not yet form the clearly metastable basins observed in BioEmu; however, whereas BioEmu achieves metastability by becoming trapped in the *holo* state, PLaTITO-19M's limitation appears to be energetic separation rather than mode coverage, indicating room for improvement through further data or model scaling.

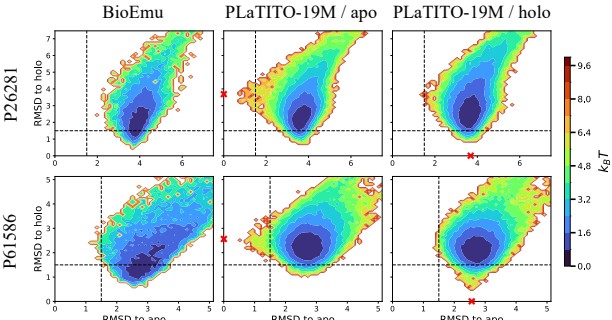

*Figure 6.* Free-energy surfaces of the local RMSD to the holo (y-axis) and apo (x-axis) reference states estimated from 100 independent trajectories sampled by PLaTITO-19M. Trajectories are initialized from the apo state (middle) and the holo state (right). PLaTITO-19M samples apo-like conformations when initialized from holo state and vice versa. However, it does not form clearly metastable basins in either case, suggesting that further data or model scaling may be required. Dashed lines denote the success threshold of 1.5 Å and red crosses indicate the initial states. Results for all four cases are shown in Appendix D.7.

## 6. Limitations

**Coarse-grained representation** To scale to proteins we have adopted a coarse-grained $C\alpha$-only representation in our proposed models. As such, the models generate coarse-grained trajectories and ignore side-chain rearrangements that may be critical for tasks such as ligand specificity or allosteric regulation, and may limit the model's ability to accurately capture thermodynamic and kinetic properties.

**Surrogate model** The models presented are variants of ITO models and thereby inherit limitations of those, including no formal guarantees of unbiased dynamics, detailed balance, semi-group self-consistency (Chapman-Kolmogorov), or stability. Further, since the model is trained on limited data, generalization across chemical space (including protein-complexes) and thermodynamic condition (temperature) will likely be limited as well.

## 7. Conclusion

In this work, we investigate the impact of various auxiliary conditioning information from pre-trained embeddings on learning transferable implicit transfer operator models of high-dimensional molecular dynamics transition probabilities. We find that, in particular, incorporating pre-trained embeddings of protein language models (pLM) provides powerful inductive biases that allow TITO models to learn transferability more efficiently from diverse off-equilibrium trajectories. Our best performing model, PLaTITO-19M, achieves state-of-the-art performance on equilibrium sampling benchmarks in an out-of-distribution setting, notably surpassing the BioEmu baseline across all metrics, on fast-folders and cryptic binding pocket sampling. Crucially, our

model achieves these results with a nearly ten-fold reduction in both training data and compute budget. This disparity highlights a central finding of our work: while Boltzmann Emulators rely on large-scale data to map approximate equilibrium densities, using the auto-correlation structure of MD data can lower the data and compute costs of learning MD surrogates and including auxiliary conditioning information can further boost generalization and performance.

Beyond stationary distributions, PLaTITO-19M qualitatively captures the kinetics of protein folding. By predicting non-Arrhenius temperature-dependent rates, the model demonstrates that it has learned physically meaningful dynamics consistent with the rugged energy landscapes of proteins observed in experimental studies. These results suggest that PLaTITO-19M has learned to approximate the underlying propagator beyond a simple exponential scaling of rates with temperature prescribed by Arrhenius.

Looking forward, several avenues for improvement remain. While PLaTITO-19M outperforms BioEmu across several benchmarks, it still fails to capture certain meta-stable phases in the fast-folders, and the *apo* structures of some of the cryptic pocket systems, and systematically underestimates time-scales. These observations suggest that future work must focus on scaling both model capacity and data and extend results to all-atom representations. Furthermore, including data with complexes, with other proteins, biomolecules or small molecule ligands could open up PLaTITO as a tool for physically grounded screening in drug-discovery campaigns and accelerate scientific discovery at a fraction of the cost of MD simulations.

## Acknowledgements

This work was partially supported by the Wallenberg AI, Autonomous Systems and Software Program (WASP) funded by the Knut and Alice Wallenberg Foundation. SO and BP thank Knut and Alice Wallenberg Foundations for a Data Driven Life Sciences grant. SO acknowledges funding from the Chalmers Academic Excellence Program. Model training and inference was made possible by an allocation on the Berzelius resource provided by the Knut and Alice Wallenberg Foundation at the National Supercomputer Centre hosted by the National Academic Infrastructure for Supercomputing in Sweden (NAISS) (project: Berzelius-2025-189), partially funded by the Swedish Research Council through grant agreement no. 2022-06725. OW and PA were in part funded by the Novo Nordisk Foundation through the Center for Basic Machine Learning Research in Life Science (NNF20OC0062606), CAZAI (NNF22OC0077058) and a Gefion supercomputer voucher grant (NNF25OC0105153). OW and PA acknowledge support from the Pioneer Center for AI, DNRF Grant Number P1.

## Impact Statement

This work provides tools for accelerating protein molecular dynamics by training data-efficient MD surrogate models. The most direct applications lie in computational biophysics and drug discovery, where PLaTITO can accelerate conformational sampling, folding kinetics analysis and cryptic binding pocket exploration at a fraction of the cost of conventional MD. The fact that the training of these models is much more data efficient than Boltzmann Emulators is also valuable to the broader scientific community, as it enables research groups to build MD surrogates without requiring large-scale computational resources.

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

# A. Architectural details

## A.1. Conditional Embeddings

**Continuous variables**: To condition on the physical time $\Delta t$, the simulation temperature $T$ and the flow-matching time $s$, we use sinusoidal positional encodings, originally introduced in (Vaswani et al., 2017).

**Categorical variables**: To condition on the amino-acid sequence $S$ (in the base TITO model when pretrained sequence embeddings are not used) and the LLM-derived metadata $A_{\text{LLM}}$, we employ nominal embeddings. Each category $c \in C$ is mapped to a continuous n-dimensional feature vector via a learned embedding function $f : C \to \mathbb{R}^n$, where $C$ denotes the set of categorical values.

## A.2. Sequence embeddings

We extract sequence embeddings for PLaTITO using the `esmc-300m-2024-12` model and for PLaTITO-19M using the `esmc-6b-2024-12` model. Since ESM models do not support sequences with non-canonical residues, we extract sequence embeddings for Villin using the experimental structure sequence (PDB entry 2F4K), where NLE (Norleucine) is replaced by LEU (Leucine).

## A.3. Structure embeddings

For PLaTITO+Struct and PLaTITO+Struct+LLM models, we extract structure embeddings using the smallest pretrained Proteina model (Geffner et al., 2025) that contains 60M transformer parameters without any triangular attention layers. Since structure embeddings are computed online during training, we extract residue representations after the second transformer layer to reduce computation overhead.

# B. Training and Sampling details

## B.1. Hyperparameters

*Table 2.* Training hyperparameters of TITO models.

|  | PLATITO[1] | PLATITO+STRUCT[2] | PLATITO-19M |
|---|---|---|---|
| **DATA** | | | |
| BATCH SIZE | 140 | 140 | 82 |
| MAX TIME-STEP ($ns$) | 200 | 200 | 200 |
| **ARCHITECTURE** | | | |
| RESIDUE REPR DIM | 128 | 128 | 256 |
| RESIDUE COND DIM | 128 | 128 | 196 |
| PAIR REPR DIM | 128 | 128 | 196 |
| # REGISTERS | 10 | 10 | 10 |
| # ATTENTION HEADS | 6 | 6 | 6 |
| # TRANSFORMER LAYERS | 3 | 3 | 6 |
| # TRAINABLE PARAMETERS | 3M | 3M | 19M |
| **TRAINING STAGE** | | | |
| OPTIMIZER | ADAMW | ADAMW | ADAMW |
| INITIAL LR | 0.001 | 0.001 | 0.0001 |
| GRADIENT CLIP | 0.1 | 0.1 | 0.1 |
| # STEPS | 400K | 350K | 200K |
| # GPUS | 8 | 8 | 8 |
| # GPU HOURS | 1100 | 1100 | 1100 |

[1] TITO was trained using the same hyperparameters as PLaTITO.
[2] PLaTITO+Struct+LLM and TITO+Struct were trained using the same hyperparameters as PLaTITO+Struct.

*Table 3.* Sampling hyperparameters for the results in the main text.

|  | FIGURE 2 | FIGURE 3 | FIGURE 4 (TOP) | FIGURE 4 (BELOW) | FIGURE 5 | FIGURE 6 |
|---|---|---|---|---|---|---|
| TIME-STEP ($ns$) | 1 | 1 | 1 | 1 | 1 | 1 |
| TEMPERATURE (K)[1] | MD | MD | MD | MD | MD | 350 |
| SIMULATION STEPS[2] | 1,000 | 1,000 | 1,000 | 120,000 | 1,000 | 1,000 |
| # TRAJECTORIES | 1,000 | 1,000 | 1,000 | 1 | 1,000 | 100 |
| ODE STEPS | 50 | 50 | 50 | 50 | 50 | 50 |
| INTEGRATOR | EULER | EULER | EULER | EULER | EULER | EULER |

[1] For each system, temperatures are set to the temperature of the corresponding reference MD except for Trp-cage (Table 4).
[2] For the NTL9 system, we generate 5,000 rollout steps instead of 1,000 to reach equilibrium. This is consistent with the original MD dataset (Lindorff-Larsen et al., 2011) where NTL9 required substantially longer simulation times than other systems to observe folding and unfolding events.

*Table 4.* Simulation temperatures (in $K$) of the fast-folding proteins used in reference MD and in TITO sampling. For Trp-cage, we increased the temperature to 310 $K$, as the reference MD temperature of 290 $K$ lies well outside the temperature range covered during training (320–450 K).

| SYSTEM | MD | TITO[1] |
|---|---|---|
| TRP-CAGE | **290** | **310** |
| BBA | 325 | 325 |
| VILLIN | 360 | 360 |
| WW DOMAIN | 360 | 360 |
| NTL9 | 355 | 355 |
| PROTEIN B | 340 | 340 |
| BBL | 298 | 298 |
| HOMEODOMAIN | 360 | 360 |
| PROTEIN G | 350 | 350 |
| A3D | 370 | 370 |
| $\lambda$-REPRESSOR | 350 | 350 |

[1] The same values were used for all TITO variants.

## B.2. BioEmu

For all comparisons with BioEmu, we sample 10,000 configurations using the default `bioemu-v1.1` checkpoint corresponding to the model weights used to produce the results in (Lewis et al., 2025). We use the default sampling hyperparameters and compute validation metrics following the official BioEmu code repository.

## B.3. Estimation of observables

To compute dynamic observables in Sections 5.3 and 5.4, we construct Markov State Models (MSMs) following the procedure of Schreiner et al. (2023). We cluster the TICA-reduced space of the MD reference trajectories using k-means and identify folded and unfolded states using PCCA (Perron Cluster-Cluster analysis) (Röblitz & Weber, 2013). Trajectories generated by PLaTITO-19M are then projected into the same TICA space and assigned to clusters using the MD-derived cluster centers. MSMs are constructed from these assignments and used to estimate dynamical observables. Reported uncertainties correspond to standard deviations obtained from Bayesian MSM posterior sampling.

## B.4. Algorithms for training and sampling

---

**Algorithm 1** TITO training using rectified CFM

---

**Input:** MD trajectories $\{(X^j, S^j, T^j)\}_{j=1}^n$
**Input:** max time-step $\Delta t_{max}$ // in ns
**Input:** conditioning network $f_c$
**Input:** velocity network $f_v$
**while** not converged **do**
    Sample $j \sim \text{Uniform}(\{1, \ldots, n\})$
    Sample $t \sim \text{Uniform}(\{1, \ldots, |X^j| - \Delta t_{\max}\})$
    Sample $\Delta t \sim \text{Uniform}(\{1, \ldots, \Delta t_{\max}\})$
    Sample $s \sim \mathcal{U}(0, 1)$
    Sample $\varepsilon \sim \mathcal{N}(0, I)$
    $x_t = X_t^j$
    $x_{t+\Delta t} = X_{t+\Delta t}^j$
    Subtract center of gravity from $x_t, x_{t+\Delta t}$
    $z_s = s\, x_{t+\Delta t} + (1 - s)\, \varepsilon$
    $c = f_c(x_t, \Delta t, S^j, T^j)$[1]
    $v_s = x_{t+\Delta t} - \varepsilon$
    Take gradient step on $\nabla_\theta \left[ \|f_v(z_s, s, c) - v_s\|^2 \right]$
**end while**
**Output:** $f_c, f_v$

---

[1] $f_c$ optionally incorporates external representations introduced in Section 4.3, depending on the model variant.

---

**Algorithm 2** Sampling from $p(x_{\Delta t} \mid x_0, \Delta t, S, T)$

---

**Input:** initial structure $x_0$
**Input:** sequence $S$, temperature $T$, time-step $\Delta t$
**Input:** conditioning network $f_c$, velocity network $f_v$
**Input:** number of ODE steps $N$
**Input:** discretization of the unit interval $0 = s_0 < s_1 < \cdots < s_N = 1$
Sample $\varepsilon \sim \mathcal{N}(0, I)$
$z_0 = \varepsilon$
Subtract center of gravity from $x_0$
$c = f_c(x_0, \Delta t, S, T)$[1]
**for** $i = 1$ **to** $N$ **do**
    $\delta_i = s_i - s_{i-1}$
    $v_i = f_v(z_{s_{i-1}}, s_{i-1}, c)$
    $z_{s_i} = z_{s_{i-1}} + \delta_i\, v_i$
**end for**
**Output:** $x_{\Delta t} = z_1$

---

[1] $f_c$ optionally incorporates additional representations introduced in Section 4.3, depending on the model variant.

---

**Algorithm 3** Iterative rollout for trajectory generation

---

    **Input:** initial structure $x_0$
    **Input:** sequence $S$, temperature $T$, time-step $\Delta t$
    **Input:** number of rollout steps $K$
    Allocate trajectory buffer $\mathcal{T} \in \mathbb{R}^{(K+1) \times \dim(x_0)}$
    $\mathcal{T}[0] = x_0$
    **for** $k = 0$ **to** $K - 1$ **do**
        Sample $x_{(k+1)\Delta t} \sim p(x_{(k+1)\Delta t} \mid x_{k\Delta t}, \Delta t, S, T)$ using Algorithm 2
        $\mathcal{T}[k + 1] = x_{(k+1)\Delta t}$
    **end for**
    **Output:** trajectory $\mathcal{T} = \{x_0, x_{\Delta t}, \ldots, x_{K\Delta t}\}$

---

## B.5. Metrics

To quantitatively evaluate the ability of our models to sample from the equilibrium distribution, we follow the evaluation protocol of BioEmu (Lewis et al., 2025). For each system, we fit a TICA model (Pérez-Hernández et al., 2013) on pairwise $C_\alpha$–$C_\alpha$ distances from the reference MD trajectory and project both reference and generated samples into the two slowest TICA components. The projected coordinates are discretized into a $50 \times 50$ grid, resulting in normalized histogram estimates $p_i^{\text{MD}}$ and $p_i^{\text{pred}}$ for each bin $i$. Relative free energies are then computed as

$$G_i = -k_B T \ln p_i + \text{const} \tag{11}$$

where $k_B$ is the Boltzmann constant and $T$ is the simulation temperature. We report three metrics, averaged over all test systems:

- **MAE**:

$$\text{MAE} = \frac{1}{|B|} \sum_{i \in B} \left| G_i^{\text{pred}} - G_i^{\text{MD}} \right| \tag{12}$$

- **RMSE**:

$$\text{RMSE} = \sqrt{\frac{1}{|B|} \sum_{i \in B} \left( G_i^{\text{pred}} - G_i^{\text{MD}} \right)^2} \tag{13}$$

- **Coverage**: Fraction of reference bins that are also covered by the generated samples.

where $B = \{i : p_i^{\text{MD}} > 0 \text{ and } p_i^{\text{pred}} > 0\}$ is the set of bins covered by both distributions.

## C. LLM-derived annotations

When sampling with models trained using LLM-derived annotations, we set $S_{LLM} = $ "Yes" and $C_{LLM} = $ "High" for all test systems.

*Table 5.* Suitability labels and associated confidence scores produced by the DeepSeek Reasoner at a sampling temperature of 0.2 for mdCATH dataset.

| SUITABILITY $S_{LLM}$ | | CONFIDENCE $C_{LLM}$ | | |
| --- | --- | --- | --- | --- |
| YES | NO | HIGH | MEDIUM | LOW |
| 2375 (44%) | 3023 (56%) | 647 (12%) | 4746 (88%) | 5 (<1%) |

*Table 6.* We compare DeepSeek Chat and DeepSeek Reasoner on a subset of 50 manually annotated domains. The Reasoner model shows higher agreement with expert labels, with remaining discrepancies largely confined to borderline cases.

| | EXPERT AGREEMENT | | DISAGREEMENT CONFIDENCE | | |
| --- | --- | --- | --- | --- | --- |
| MODEL | YES | NO | HIGH | MEDIUM | LOW |
| CHAT | 29 (58%) | 21 (42%) | 1 | 20 | 0 |
| REASONER | 47 (94%) | 3 (6%) | 1 | 2 | 0 |

An example prompt follows:

```
SYSTEM ROLE:
You are a structural bioinformatician specializing in molecular dynamics (MD) simulations.

TASK:
You are tasked to screen a dataset of protein domain annotations to determine whether the
protein domains are likely to be useful for molecular dynamics simulations. In particular,
you are tasked to formulate a "yes" or "no" opinion, provide a confidence level, explain
which evidence it is based on. The only information available to formulate your opinion is
the annotations in the dataset. Expect some empty columns. Assume the simulations are done
in standard conditions (1 atm, 300 K, NVT ensemble, TIP3P water, Na+ and Cl- ions
at 0.150 M concentration to charge neutralize the systems), and all non-protein atoms
have been stripped from the systems during preprocessing.

[CRITERIA]
Consider all the annotations, but especially:
- the completeness of the protein domain annotation
- cath_domain_length, the length of the protein domain
- full_chain_domain, if the protein domain is a full chain
- pdb_polymer_composition, the composition of the assembly in the PDB entry,
possibly molecules interacting with the protein domain
- pdb_assembly_count, usually the same assembly but with different symmetry
- pdb_entity_count, the number of molecules in the PDB entry
...
- organism
- protein_name
- cc_subcell_location
...

[DOMAIN KNOWLEDGE]
Some key considerations:
- long domains (cath_domain_length) might be more stable
and useful for MD simulations, but also more complex
```

- full chain domains (full_chain_domain) are more likely
to be stable and useful for MD simulations
- cc_subcell_location or organism could hint at the pH,
temperature, and ionic strength of the environment,
which could affect the stability of the protein domain
...

[OUTPUT FORMAT]
Reply with a JSON object containing exclusively the following fields:
- "CATH ID": the value of the "cath_id" column in the input data
- "Classification": "Yes" or "No"
- "Confidence": "High", "Medium", or "Low"
- "Evidence": a direct quotation of the most relevant annotations in the input data

[DATA]
Here is the annotation data for this protein domain:

# D. Additional results

## D.1. Equilibrium sampling - Ablation of TITO variants

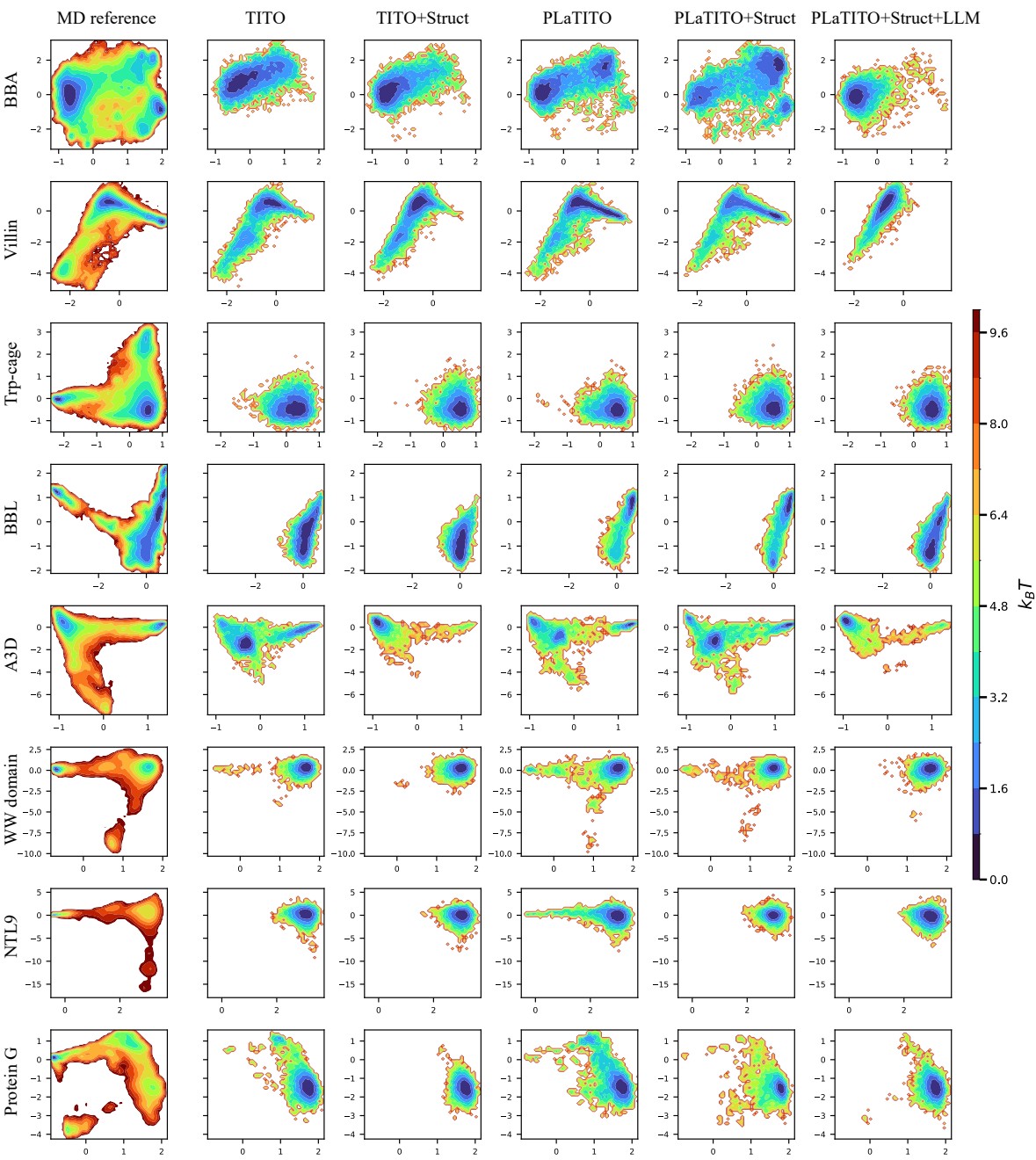

*Figure 7.* Free energy surfaces projected into the two slowest TICA components of BBA, Villin, Trp-cage, BBL, A3D, WW domain, NTL9 and Protein G sampled by our proposed TITO models compared to the MD reference distributions (left).

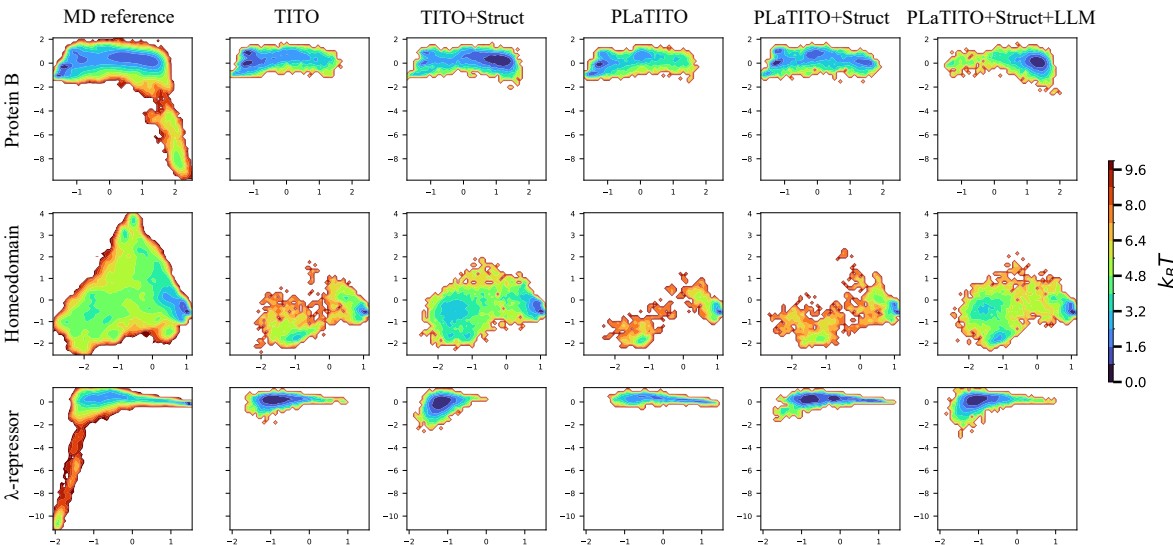

*Figure 8.* Free energy surfaces projected into the two slowest TICA components of Protein B, Homeodomain and $\lambda$-repressor sampled by our proposed TITO models compared to the MD reference distributions (left)

## D.2. Equilibrium sampling - Comparison with BioEmu

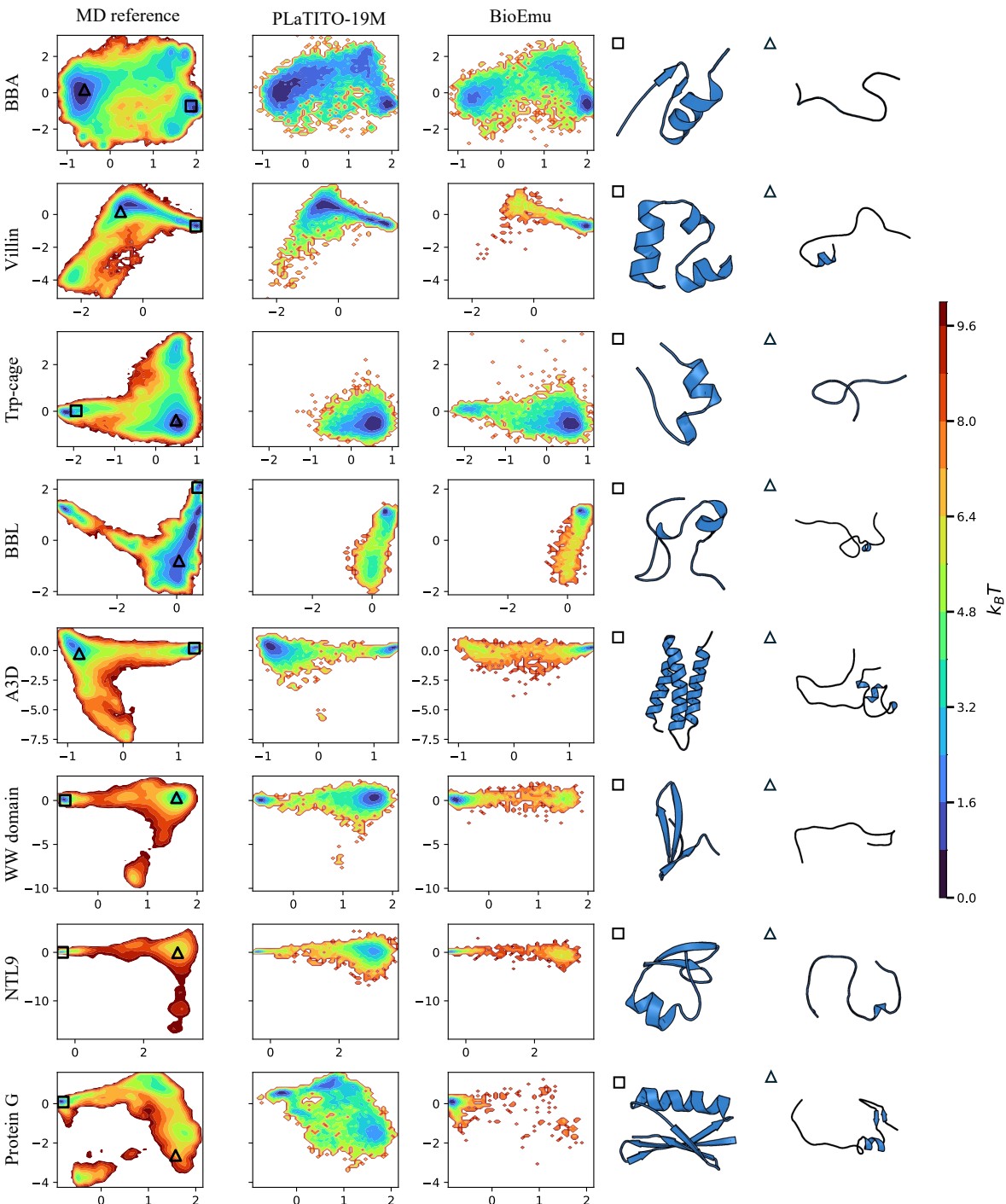

*Figure 9.* Free energy surfaces projected into the two slowest TICA components of BBA, Villin, Trp-cage, BBL, A3D, WW domain, NTL9 and Protein G sampled by PLaTITO-19M (middle) and compared to the MD reference distributions (left) and BioEmu (right). Squares (□) and triangles (△) denote folded and unfolded states, respectively, with PLaTITO-19M trajectories initialized from the unfolded state.

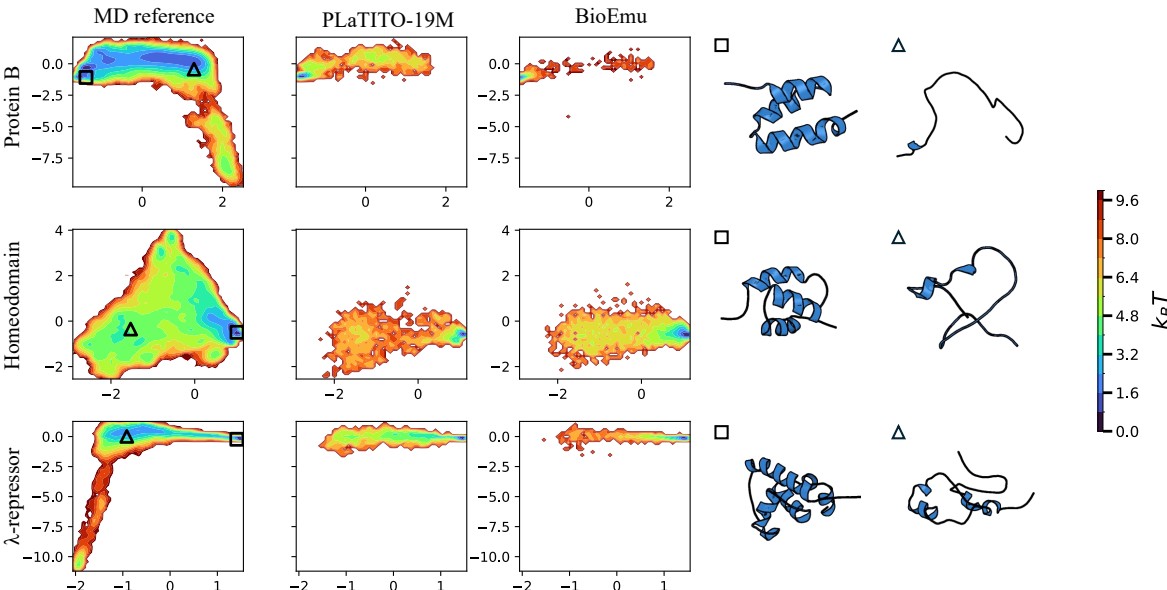

*Figure 10.* Free energy surfaces projected into the two slowest TICA components of Protein B, Homeodomain and $\lambda$-repressor sampled by PLaTITO-19M (middle) and compared to the MD reference distributions (left) and BioEmu (right). Squares ($\square$) and triangles ($\triangle$) denote folded and unfolded states, respectively, with PLaTITO-19M trajectories initialized from the unfolded state.

### D.3. Equilibrium sampling - Ablation of pretrained pLMs

To disentangle the contribution of the protein language model embeddings from the specific choice of pLM model and size, we conduct an ablation where we change the pretrained pLM used to extract sequence embeddings while keeping all other parameters of the model fixed. We consider models from two ESM families: ESMC at three scales (300M, 600M, and 6B parameters) (ESM Team, 2024) and ESM2 (650M parameters) (Lin et al., 2023).

*Table 7.* Equilibrium sampling performance when changing pLM at fixed model capacity (3M parameters)

| Family | Size | Dimension | MAE ($\downarrow$) | RMSE ($\downarrow$) | Coverage ($\uparrow$) |
|--------|------|-----------|------|------|----------|
| - | - | - | 1.068±0.272 | 1.382±0.302 | 0.590±0.111 |
| ESMC | 300M | 960 | 0.949±0.269 | 1.228±0.328 | 0.651±0.151 |
| ESMC | 600M | 1152 | 0.964±0.268 | 1.307±0.303 | 0.578±0.072 |
| ESMC | 6B | 2560 | 0.976±0.251 | 1.261±0.268 | 0.611±0.080 |
| ESM2 | 650M | 1280 | 0.968±0.261 | 1.256±0.290 | 0.579±0.090 |

*Table 8.* Equilibrium sampling performance when changing pLM at fixed model capacity (19M parameters)

| Family | Size | Dim | MAE ($\downarrow$) | RMSE ($\downarrow$) | Coverage ($\uparrow$) |
|--------|------|-----|------|------|----------|
| ESMC | 300M | 960 | 0.840±0.228 | 1.124±0.262 | 0.674±0.105 |
| ESMC | 6B | 2560 | 0.824±0.170 | 1.099±0.212 | 0.666±0.136 |

In Table 7, we report results for PLaTITO-3M and in Table 8 for PLaTITO-19M. In all cases, incorporating sequence embeddings from pLMs consistently improves generalization over the base TITO model, further strengthening our findings that pLM representations encode useful thermodynamic information. Interestingly, the benefit of larger pLMs depends on model capacity. In PLaTITO-3M, the smaller ESMC-300M performs best, suggesting that PLaTITO-3M model lacks sufficient capacity to fully exploit the higher-dimensional embeddings of ESMC-6B. When scaling to PLaTITO-19M, we see that the larger embeddings from ESMC-6B yield better MAE and RMSE. This indicates that scaling pLM representations and model capacity are complementary since higher-dimensional embeddings require increased model capacity to be fully leveraged.

### D.4. Equilibrium sampling - Ablation of model capacity

To investigate the effect of model capacity independently of the pLM choice, we train PLaTITO variants ranging from 1M to 100M parameters while keeping the pLM fixed to ESMC-6B (Table 9). All models are trained using the same dataset split and training compute budget.

*Table 9.* Equilibrium sampling performance when scaling PLaTITO parameters at a fixed pLM (ESMC-6B).

| Parameters | MAE ($\downarrow$) | RMSE ($\downarrow$) | Coverage ($\uparrow$) |
|------------|------|------|----------|
| 1M | 1.253±0.366 | 1.605±0.459 | 0.569±0.101 |
| 3M | 0.976±0.251 | 1.261±0.268 | 0.611±0.080 |
| 19M | 0.824±0.170 | 1.099±0.212 | 0.666±0.136 |
| 36M | **0.811**±0.213 | **1.066**±0.252 | 0.664±0.111 |
| 100M | 0.858±0.266 | 1.115±0.280 | **0.668**±0.115 |

We observe that performance improves consistently from 1M to 36M parameters, with a slight degradation at 100M. This is consistent with scaling laws where further capacity scaling requires a corresponding increase in training data.

### D.5. Equilibrium sampling - Comparison with additional baselines

Apart from BioEmu (Lewis et al., 2025), we compare PLaTITO against two additional models that report results on equilibrium sampling of the fast-folding proteins. ConfDiff (Wang et al., 2024) is a force-guided SE(3) diffusion model that

incorporates MD energy priors to guide conformation generation towards the Boltzmann distribution while Str2Str (Lu et al., 2024) is a score-based structure-to-structure translation framework for zero-shot protein conformation sampling.

In Table 10, we observe that PLaTITO-19M achieves the best MAE and RMSE by a substantial margin. While Str2Str achieves the highest Coverage, this metric alone can be misleading since a model that generates diverse but unrealistic samples will cover many bins while poorly approximating the actual free energy landscape. This is exactly what we observe in the respective TICA plots (Figures 11 and 12), where Str2Str generates unrealistic metastable basins.

*Table 10.* Performance on equilibrium sampling of the fast-folding proteins across different models.

| Model | MAE (↓) | RMSE (↓) | Coverage (↑) |
|---|---|---|---|
| ConfDiff | 1.419±0.200 | 1.709±0.264 | 0.535±0.125 |
| Str2Str | 0.998±0.262 | 1.274±0.293 | **0.707**±0.161 |
| BioEmu | 1.110±0.292 | 1.389±0.346 | 0.594±0.175 |
| PLaTITO-19M | **0.824**±0.170 | **1.099**±0.212 | 0.666±0.136 |

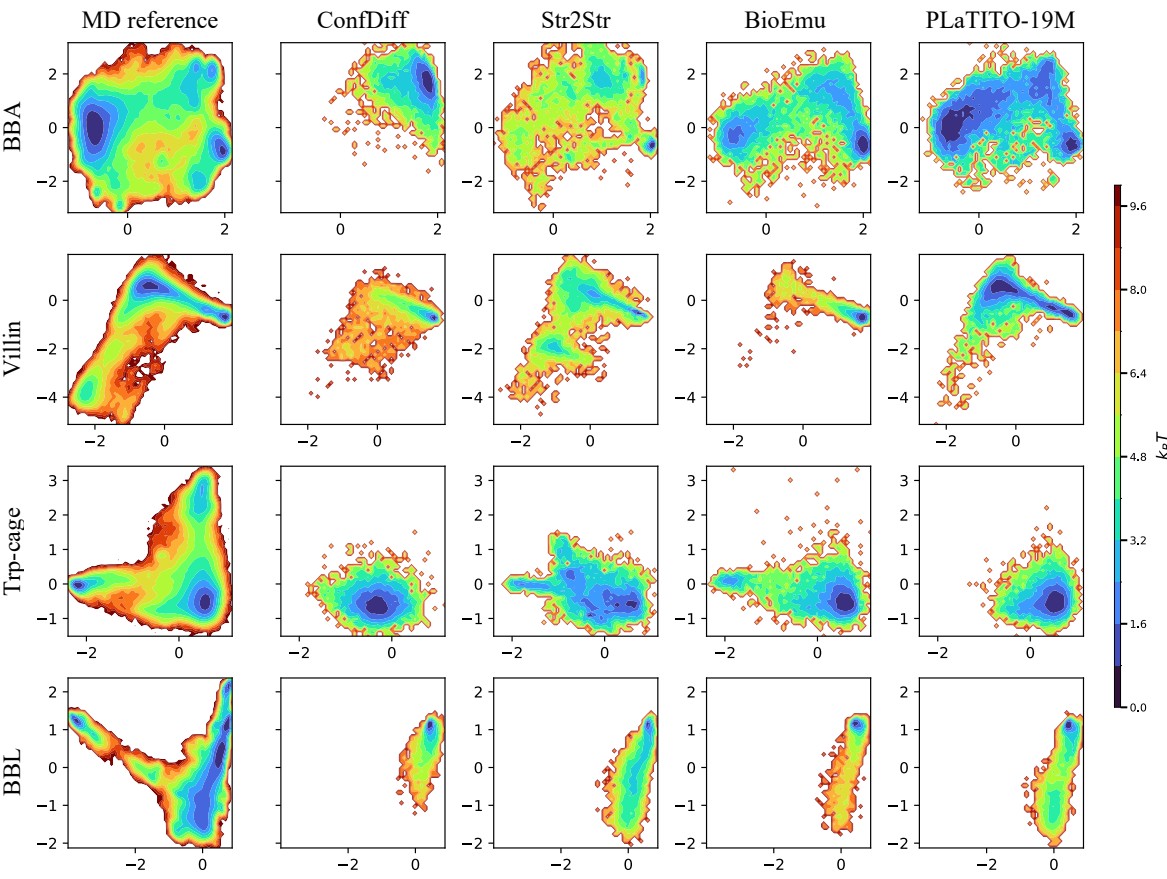

*Figure 11.* Free energy surfaces projected into the two slowest TICA components of BBA, Villin, Trp-cage and BBL sampled by different models and compared to the MD reference distributions.

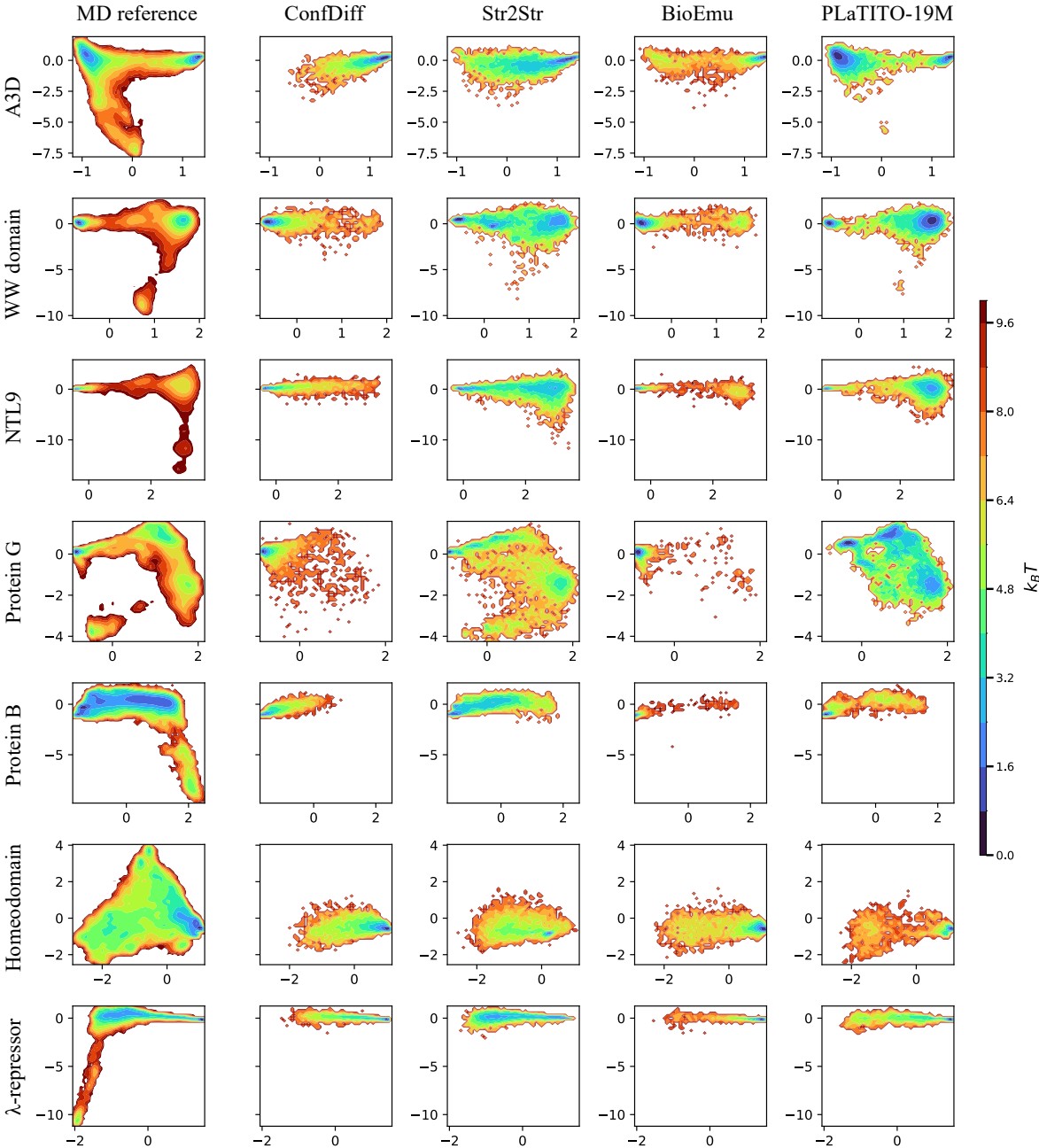

*Figure 12.* Free energy surfaces projected into the two slowest TICA components of A3D, WW domain, NTL9, Protein G, Protein B, Homeodomain and $\lambda$-repressor sampled by different models and compared to the MD reference distributions.

## D.6. Long time-step dynamics

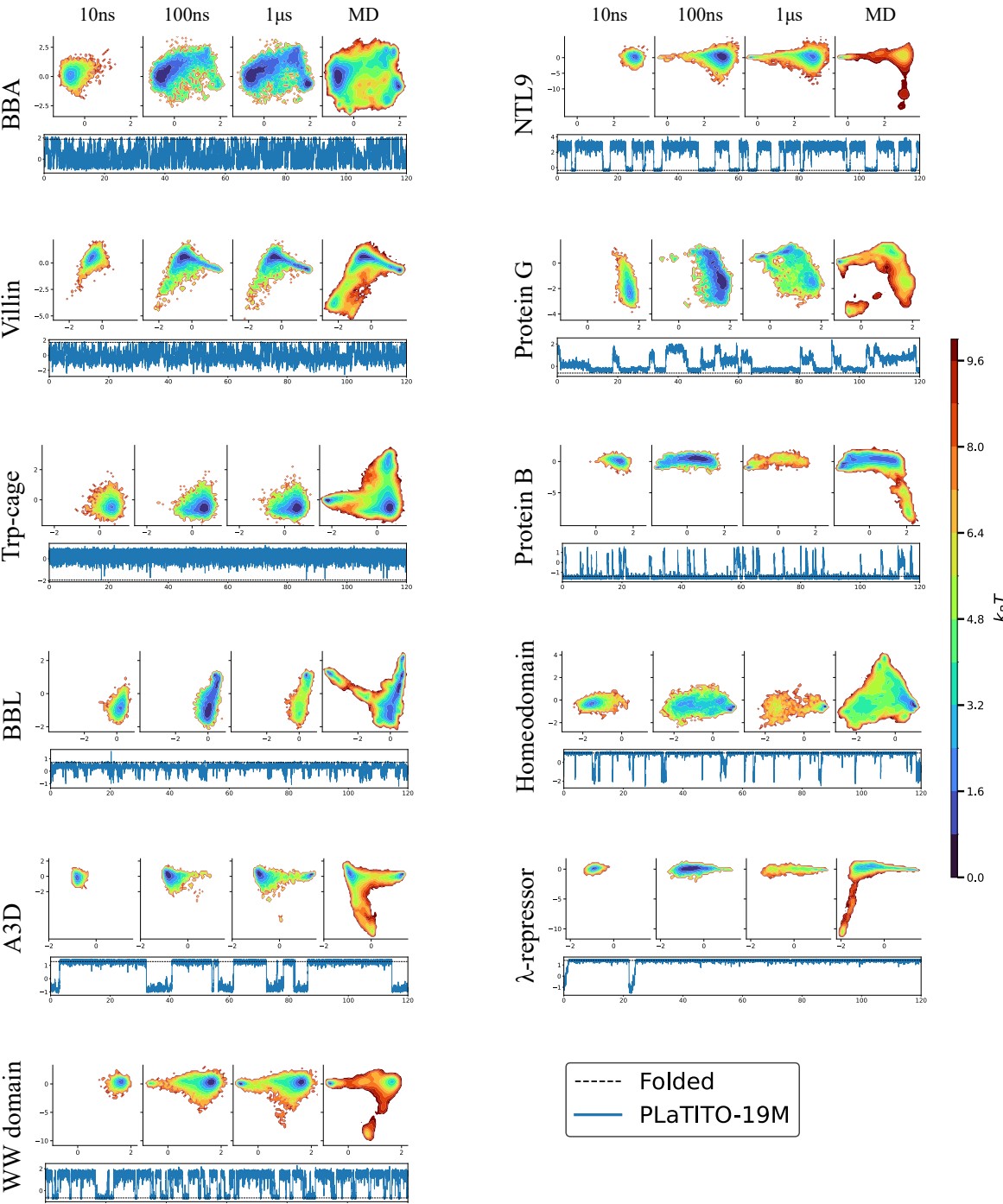

*Figure 13.* Free-energy surfaces projected into the two slowest TICA components for multiple fast-folding proteins, estimated by PLaTITO-19M from 1,000 independent trajectories initialized from unfolded conformations. Distributions are shown at increasing rollout times (10 ns, 100 ns, and 1 μs, left to right) and compared to the MD reference distributions (rightmost column). For each system, a representative long trajectory by PLaTITO-19M of 120 μs projected in the slowest TICA component is shown below, illustrating repeated folding and unfolding events.

## D.7. Formation of cryptic binding pockets

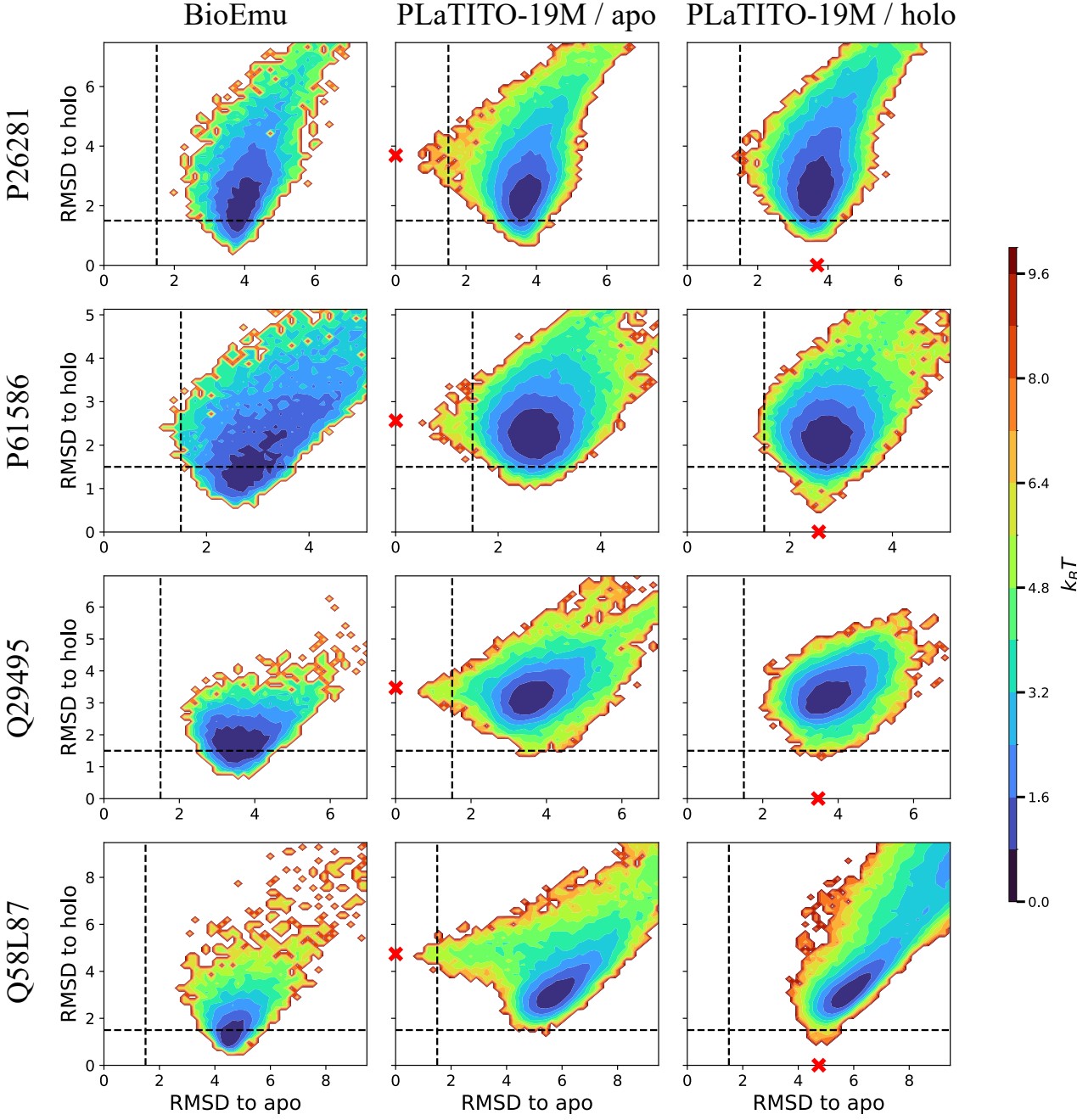

*Figure 14.* Free-energy surfaces of the local RMSD to the holo (y-axis) and apo (x-axis) reference states estimated from 100 independent trajectories sampled by PLaTITO-19M. Trajectories are initialized from the apo state (middle) and the holo state (right). Dashed lines denote the success threshold of 1.5 Å and red crosses indicate the initial states.

# E. Inference speed

We evaluate the inference speed of PLaTITO-19M by measuring how many simulation steps can be sampled per second on a single NVIDIA H100 GPU. For each system, we determine the maximum number of parallel trajectories that can fit in the memory and report the average throughput in simulation steps per second.

*Table 11.* Inference speed of PLaTITO-19M measured on a single NVIDIA H100 GPU.

| SYSTEM | # RESIDUES | # PARALLEL TRAJECTORIES | SIMULATION STEPS/SEC |
| --- | --- | --- | --- |
| TRP-CAGE | 20 | 16,000 | 696 |
| BBA | 28 | 12,000 | 450 |
| VILLIN | 35 | 8,000 | 318 |
| WW DOMAIN | 35 | 8,000 | 318 |
| NTL9 | 39 | 7,000 | 273 |
| PROTEIN B | 47 | 5,000 | 201 |
| BBL | 47 | 5,000 | 201 |
| HOMEODOMAIN | 52 | 4,000 | 172 |
| PROTEIN G | 56 | 4,000 | 162 |
| A3D | 73 | 2,000 | 97 |
| $\lambda$-REPRESSOR | 80 | 2,000 | 87 |

We also leverage the PyTorch compilation framework (Ansel et al., 2024) to speed up inference. Using compiled models, the inference throughput of the largest system, $\lambda$-repressor, increases from 44 to 87 simulation steps per second compared to uncompiled execution.

