# OpenReview forum: "Protein Language Model Embeddings Improve Generalization of Implicit Transfer Operators"
_ICML.cc/2026/Conference — ICML 2026 regular_

### Official Review · Reviewer_xcH3 · 2026-03-04

**Soundness:** 2
**Presentation:** 3
**Significance:** 2
**Originality:** 3
**Overall Recommendation:** 4
**Confidence:** 3

**Summary:**

The authors propose a novel MCMC method for protein ensemble generation or MD simulation acceleration. The approach utilizes various conditioning types, including protein language model (PLM) embeddings and text LLM-derived annotations (e.g., from DeepSeek R1). The proposed PLaTITO model is trained and evaluated on the mdCATH dataset.

**Compliance With Llm Reviewing Policy:**

Affirmed.

**Final Justification:**

I have comprehensively assessed the author's rebuttal and the additional results. I have increased my evaluation accordingly. Although the authors have addressed my concerns in the questions, the overall contribution (methodological, experimental) of this paper is limited and lack of wide application to larger protein system.

**Key Questions For Authors:**

1. How do you justify the claimed “out-of-distribution” behavior as mentioned in the main contribution paragraphs? Could the author provide additional details how the test set differs from the training distribution in which regard? While the test set contains at most 40% sequence similarity but one should be careful and require specific evidence to make this claim.

2. Missing other published baselines: MCMC baselines (eg. TimeWarp[1]), baselines trained on mdCATH (eg., TEMPO [2]), and baselines evaluated on the fast folders (eg., Str2Str [3], ConfDiff[4]).

3. From Table 1/Figure 7, it seems incorporating the LLM annotations basically hurts the performance and visualized distribution coverage. Do the authors want to further justify the necessity with additioanl ablation study to support this conditioning, which can be an important innovation of the proposed method?

4. Could the authors explain what is the “non-Arrhenius temperature-dependent kinetics learned by PLaTITO”?

5. The model is trained on mdCATH trajectories where the simulation on average lasts hundreds of nanosecond, according to section 4.4. How does this training enable the correct sampling of fast folders ms-level distribution? Or is it just a random guess?


6. What is the largest protein system (w.r.t. number of residues or Ca atoms) during (1) training and (2) inference stage?

[1] Klein, Leon, Andrew Foong, Tor Fjelde, Bruno Mlodozeniec, Marc Brockschmidt, Sebastian Nowozin, Frank Noé, and Ryota Tomioka. "Timewarp: Transferable acceleration of molecular dynamics by learning time-coarsened dynamics."

[2] Xu, Yaoyao, Di Wang, Zihan Zhou, Tianshu Yu, and Mingchen Chen. "TEMPO: Temporal Multi-scale Autoregressive Generation of Protein Conformational Ensembles."

[3] Lu, Jiarui, Bozitao Zhong, Zuobai Zhang, and Jian Tang. "Str2str: A score-based framework for zero-shot protein conformation sampling."

[4] Wang, Yan, Lihao Wang, Yuning Shen, Yiqun Wang, Huizhuo Yuan, Yue Wu, and Quanquan Gu. "Protein conformation generation via force-guided se (3) diffusion models."

**Limitations:**

Yes

**Strengths And Weaknesses:**

Pros:

  1. The authors innovatively use LLM annotations for the condition.
2. The work successfully scales up the ITO models to general proteins.

Cons:
  1. The method is not evaluated on the standard ATLAS or mdCATH test sets as a benchmark.
  2. There is a limited baseline comparison, making the results less significant.
  3. The paper presents limited technical (in terms of modeling or architectural) contributions.
  4. The PLaTITO model only generates Ca conformation ensembles.

---

> ### Author Rebuttal · Authors · 2026-03-31
>
> We thank the reviewer for the constructive review.
>
> &nbsp;
>
> **Evaluation benchmarks**
>
> - We respectfully disagree that evaluating on ATLAS or mdCATH would strengthen the paper. Evaluation on short trajectories would constitute a weaker test since **these trajectories are far too short to have converged equilibrium distributions or reliable kinetic estimates**. Evaluating a model which aims to reproduce the thermodynamics and kinetics across the conformational space, against unconverged reference data would be scientifically misleading, since the reference itself would be unconverged. Previous work on TITO (Diez et al., 2025a, Figure 3) demonstrated **how short reference trajectories lead to unreliable evaluation**. On the other hand, fast-folders provide ms-scale reference trajectories with well-characterized equilibrium distributions and folding kinetics.
>
> - To test our claim, we sampled long trajectories of 1$\mu s$ using TEMPO (Xu et al., 2024), one of the baselines that has been evaluated only on short trajectories. In contrast to PLaTITO-Big, TEMPO completely fails to recover any meaningful equilibrium distribution and generates structures completely off from the MD reference (tempo_on_fast_folders.png in https://anonymous.4open.science/r/icml26_rebuttal-534A/) **suggesting that strong performance on short-trajectory sampling does not translate to the ability to generate converged equilibrium distributions**.
>
> &nbsp;
>
> **On out-of-distribution claim**
>
> We decreased the similarity threshold to 10% and only 2 additional domains out of the total 4,482 were removed, **confirming that the sequence similarity between mdCATH and fast folders is very low**. Also, fast-folders originate from a different study (Lindorff-Larsen et al., 2011) than mdCATH (Mirarchi et al., 2024) where different system preparation and equilibration protocols were used.
>
> &nbsp;
>
> **Missing baselines**
>
> We have benchmarked more baselines on fast folders:
>
> | Model | MAE (↓) | RMSE (↓) | Coverage (↑) |
> |-------|---------|----------|--------------|
> | ConfDiff | 1.419±0.200 | 1.709±0.264 | 0.535±0.125 |
> | Str2Str | 0.998±0.262 | 1.274±0.293 | **0.707**±0.161 |
> | BioEmu | 1.110±0.292 | 1.389±0.346 | 0.594±0.175 |
> | PLaTITO-19M | **0.824**±0.170 | **1.099**±0.212 | 0.666±0.136 |
>
> - **PLaTITO-19M achieves the best MAE and RMSE by a large margin**. While Str2Str achieves higher Coverage, this metric alone can be misleading since a model that generates diverse but unrealistic samples will cover many bins while poorly approximating the actual free energy landscape. This is exactly what we observe in the respective TICA plots (comparison_w_baselines.png in https://anonymous.4open.science/r/icml26_rebuttal-534A/), where Str2Str generates unrealistic metastable basins.
> - TimeWarp is limited to tetrapeptides with system-specific training and is not applicable.
> - TEMPO, as shown above, fails to generate stable long time-scale trajectories.
>
> &nbsp;
>
> **LLM-derived annotations**
>
> We refer to our response to Reviewer DyQi.
>
> &nbsp;
>
> **Temperature-dependent kinetics**
>
> - Arrhenius equation describes the exponential dependence of the rate on the temperature as follows: $k(T) = A·exp(−\frac{E_a}{k_BT})$ that yields a straight line when plotting log(k) vs 1/T.
> - However, protein folding is characterized by complex high-dimensional free energy landscapes populated with many transient intermediate structures (Scalley & Baker, 1997; Kragelund et al., 1999) and we expect deviation from the idealized exponential temperature dependence.
> - We show (Figure 5) that the folding and unfolding timescales predicted by PLaTITO-Big as a function of inverse temperature **exhibit a clear curvature rather than a straight line, indicating that the model has learned non-Arrhenius temperature-dependent kinetics**.
>
> &nbsp;
>
> **How training on short trajectories enables long-timescale sampling**
>
> - PLaTITO is trained to approximate $p(x_{t+\Delta t} | x_t, \Delta t, S, T)$ from pairs of structures randomly sampled at time lags $\Delta t \in [1, 200] ns$.
> - Since mdCATH contains short trajectories (~ 500 ns) we can sample pairs from these trajectories during training.
> - During inference, **to enable long-timescale sampling we apply the learned transition operator iteratively (Algorithm 3, Appendix B.4) where each step generates a new structure conditioned on the previous one**.
> - For example, to end up with 1 $\mu s$ trajectories (that are far longer than any individual training trajectory) we generate a chain of 1,000 steps with $\Delta t=1 ns$. This is the same principle by which Markov state models predict millisecond-scale kinetics from ensembles of short simulations (Prinz et al., 2011; Nüske et al., 2014).
>
> &nbsp;
>
> **Largest systems**
>
> - Training: cut-off at 200 residues (Section 4.4).
> - Inference:
>    - Cryptic pockets: P61586 (179 residues)
>    - Fast folders: $\lambda$-repressor (80 residues).
>
> &nbsp;
>
> We hope our clarifications address the reviewer’s concerns.

---

> > ### Author Rebuttal · Reviewer_xcH3 · 2026-04-04
> >
> > Thank the authors for their response. I will increase my score accordingly.

---

### Official Review · Reviewer_DyQi · 2026-03-11

**Soundness:** 2
**Presentation:** 3
**Significance:** 1
**Originality:** 1
**Overall Recommendation:** 3
**Confidence:** 5

**Summary:**

The paper presents PLaTITO, a framework that scales Implicit Transfer Operators (ITO) for protein molecular dynamics by incorporating auxiliary representations. The authors leverage pretrained protein language model (PLM) embeddings, structural embeddings, and LLM-derived metadata to condition a flow-matching-based transition model. Trained on the mdCATH dataset, the model demonstrates the ability to generalize to out-of-distribution (OOD) fast-folding proteins without system-specific fine-tuning. A key highlight is its superior data efficiency, achieving state-of-the-art equilibrium sampling and recovering non-Arrhenius kinetics with nearly an order of magnitude less compute and data than current Boltzmann emulators like BioEmu.

**Compliance With Llm Reviewing Policy:**

Affirmed.

**Key Questions For Authors:**

1. What do you believe is the primary cause of performance degradation from LLM-derived annotations, and have you explored alternative annotation strategies (e.g., continuous scores instead of binary labels) to improve results?
2. What are the main technical challenges for extending PLaTITO to support backbone representations, and do you have plans for such extensions?

**Limitations:**

Yes.

**Strengths And Weaknesses:**

### Soundness
Strengths:

The empirical validation on fast-folding proteins is rigorous, using standard TICA projections and free-energy landscape comparisons.

### Presentation
Strengths:

The paper is exceptionally well-structured. Figure 1 provides a clear overview of the multi-source conditioning architecture.

### Significance
Weaknesses:

While comparison with BioEmu is thorough, the lack of all-atom details due to the $C_\alpha$ representation limits the model's utility. It is worth noting that BioEmu is backbone-level.

### Originality
Weaknesses:
1. The main framework has been fully established. The paper explores whether various features prove useful. Incorporating PLM embedding will undoubtedly enhance performance—this is the community consensus. Consequently, the author does not provide any particularly valuable insights.
2. If works like F$^3$low [1] have already mastered backbone-level transitions, why are we regressing to $C_\alpha$ representation? It feels like a step backward.

>[1] Li S, Wang Y, Li M, et al. F$^3$low: Frame-to-Frame Coarse-grained Molecular Dynamics with SE (3) Guided Flow Matching[J]. arXiv preprint arXiv:2405.00751, 2024.

---

> ### Author Rebuttal · Authors · 2026-03-31
>
> We thank the reviewer for the constructive feedback and we address each point below.
>
> &nbsp;
>
> **Core contribution**
>
> We respectfully disagree with the claim that "incorporating pLM embeddings will undoubtedly enhance performance and it is community consensus”. While pLM embeddings have proved useful in protein structure prediction, inverse folding and fitness prediction tasks, to our knowledge, **there is no prior work demonstrating the effect on incorporating pLM embeddings in GenMD**. Learning protein dynamics is fundamentally different from structure prediction since the model has to capture the full distribution of conformational states across multiple timescales and temperatures. The fact that pLM embeddings provide useful inductive bias for this task is not at all obvious a priori and **demonstrating that they enable state-of-the-art OOD generalization with around 10× less data and compute is precisely the contribution of this paper**.
>
> &nbsp;
>
> **Backbone-level transitions**
>
> We respectfully disagree with the claim that works like F$^3$low (Li et al., 2024) have already mastered backbone-level transitions, making our $C_{\alpha}$ approach a step backward. To our knowledge, no existing method accurately generates protein conformational ensembles using backbone-level transitions in an unseen setting.
> - F$^3$low is trained and evaluated on individual protein systems and **it does not generalize to unseen proteins without system-specific retraining**. This is a fundamentally easier task than what PLaTITO addresses: zero-shot generalization to out-of-distribution protein systems trained from diverse off-equilibrium trajectories.
> - BioEmu is indeed a backbone-level model and the fact that $C_{\alpha}$-only PLaTITO outperforms it on equilibrium sampling benchmarks demonstrates that **accurate sampling does not necessarily require backbone-level resolution**.
>
> &nbsp;
>
> **Extending PLaTITO to fine-grained representations**
>
> We want to clarify that, prior to this work, **TITO for protein molecular dynamics had not been demonstrated**. Our goal was to first establish that ITO learning can generalize to out-of-distribution protein systems, which we have shown at the $C_{\alpha}$ level, outperforming the current state of the art (BioEmu) with an order of magnitude less data and compute.
>
> - Working towards a backbone-level PLaTITO is relatively straightforward since the current architecture can be extended to account for both the position and the orientation of each residue, similar to BioEmu.
> - Extending to an all-atom representation presents more challenges:
>    1. Design an architecture that effectively handles side-chain rearrangements while remaining scalable
>    2. Handle the substantially increased computational cost for training at higher resolution.
>
> However, given the earlier results on ITO illustrating that C$_{\alpha}$-CG representations are sufficient to learn faithful kinetics and thermodynamics, **it was essential to first establish whether this result carried over to (a) generalization to unseen protein systems, and (b) pLM conditioning can significantly improve data efficiency, a property that will be even more critical in the all-atom setting**.
>
> &nbsp;
>
> **LLM-derived annotations**
>
> - We incorporated LLM annotations to flag unreliable data samples motivated by results on label noise making generalization more challenging (Arpit et al. ICML 2017; Zhang et al. ICLR 2017). MD datasets contain simulations of protein domains extracted from larger complexes and many such domains naturally require binding partners or cofactors to remain stable. Simulating them in isolation can lead to unphysical behavior. **LLM annotations were designed to flag such unreliable training samples without requiring manual curation at scale.**
>
> - To investigate the underperformance, we trained another variant using an additional “evidence” variable that corresponds to the reasoning that the LLM provided alongside its annotations. However, this variant also failed to improve performance, **indicating that the issue lies in the quality of the annotations themselves**, not in how they are represented.
>
> - We then found many cases where domains received opposite suitability labels with similar justifications. For instance, 4jk8B01 and 4kooB00 were classified as "No" and "Yes" respectively, with the LLM citing essentially the same structural characteristics in both cases (homomeric protein with metallic bonds, no covalent bonds, and mutations affecting DNA/RNA interaction). **This indicates that our LLM annotations introduced substantial label noise during training rather than providing useful biological context**.
>
> - Future work could incorporate richer biological context into the prompts or leverage LLMs specifically fine-tuned for biomedical reasoning to produce more reliable annotations.
>
> &nbsp;
>
> We hope that these clarifications, along with our new ablations across all our responses will address the reviewer’s concerns.

---

### Official Review · Reviewer_1FUJ · 2026-03-13

**Soundness:** 2
**Presentation:** 3
**Significance:** 3
**Originality:** 2
**Overall Recommendation:** 5
**Confidence:** 4

**Summary:**

This paper introduces PLaTITO, which extends on the TITO framework to learn the long time transition densities of coarse-grained protein molecular dynamics, conditioned on sequence and simulation temperature, with optional conditioning inputs from pLM embeddings, structure embeddings, and LLM annotations. This type of extra conditioning improves equilibrium sampling in OOD settings relative to the base TITO model, and a larger variant of the PLaTITO outperforms the reported BioEmu baseline on the fast-folding protein benchmark while using less training data from MD and GPU time. PLaTITO is also evaluated on folding dynamics, temperature-dependent kinetics, and cryptic pocket benchmarks.

**Compliance With Llm Reviewing Policy:**

Affirmed.

**Final Justification:**

The paper presents strong empirical results, clear practical value, and meaningful advances in data-efficient protein MD modeling, despite incremental methodological novelty. The work is reasonably sound, and while some evaluation gaps remain (e.g., quantitative validation of temperature-dependent kinetics), the rebuttal clarified claims, corrected errors, and appropriately scoped contributions, which addressed my main concerns. The responses on MFPT evaluation limitations improved my confidence, even if not all issues were fully resolved experimentally. Clarity is solid and will benefit further from the proposed revisions. Overall, the rebuttal strengthened my assessment and justified raising my score to accept (5).

**Key Questions For Authors:**

I think the empirical results are very strong. I am asking a few clarifying questions to better understand the model and its limitations. I will increase the score if the authors provide suitable clarifications.

1. For A3D, the reported MFPTs are much faster than MD (Section 5.3), and this trend is also observed in BBA and Villin (Section 5.4). While authors explain that it is from the variational principle with imperfect approximations, it would be nice if the authors could disentangle the effects from (1) imperfect training of the transfer operator and (2) non-Markovian behavior from coarse-graining not captured by modeling it as a Markovian dynamics via the transfer operator. If it is primarily the latter, are there promising next steps that could fix this behavior?
2. Similar to the training compute scaling experiment in Figure 3, could the authors provide a scaling experiment in terms of model sizes? While it is impressive that 3M/19M PLaTITO models are performing as well as or better than the 31M BioEmu model, it would be nice to understand the limits of parameter efficiency enabled by the autocorrelative modeling scheme (conditioning on the previous frame).
3. Regarding the temperature trend of kinetics, could the authors perform MD simulations using the same force field settings to obtain the reference data points for MFPT? I think BBA/Villin unfolding timescales would be reachable with unbiased MD during the rebuttal period, or biased MFPT simulations, such as infrequent metadynamics/flooding simulations, could be useful as well.
4. Trp-cage was modeled at a higher temperature with PLaTITO due to the MD temperature being outside the training temperature range. How does the model behave when extrapolating the temperature range?
5. Minor points
	 - Table 1 lists the same RMSD and coverage for PLaTITO + Struct and BioEmu. Is it actually the same or a typo?
	- CHARMM22 -> CHARMM22* (line 297-298)
	- Fast folding protein dataset contains 12 proteins, but Table 4 lists 11 proteins without chignolin. Is chignolin omitted intentionally?

**Limitations:**

yes

**Strengths And Weaknesses:**

### Strengths

- PLaTITO scales effectively and achieves SOTA performance on equilibrium protein MD sampling benchmarks. Impressively, it even outperforms large-scale foundational sampling models such as BioEmu, which is trained with much more extensive computational resources.
- The design choice to integrate pLM embeddings alongside the previous time step structure is sensible considering that the sequence encodes dynamics, and it is substantiated by benchmark results. The split at 40% sequence similarity ensures the benchmarks are correctly testing the transferability of the model.
- Exposition in the paper is generally clear in explaining and motivating the model variants, and the results shown in figures and tables highlight the strengths of the model.

### Weaknesses

- While the benchmark results are quite strong, the paper is framed in a way that methodological contribution is incremental: it scales the ITO/TITO line of work to coarse-grained proteins, using a non-equivariant backbone model from Proteina, and adds pretrained conditioning from pLM/structure models. As cited in the related work section, DeepJump is similarly trained on mdCATH and tested on transferability to fast-folding proteins. While the empirical finding that pLM/structure embeddings help TITO is nice, it is relatively well-known in the literature on generative modeling for protein structures.
- The introduction of embeddings from LLM-derived annotations is interesting, but it is not clear why it could help the model learn better from annotated confidence/suitability. Also, it could be hard to interpret because the test-time annotations are set to Yes/High for all test systems.
- For some benchmarks, a quantitative match in the trend is missing. For example, in temperature-dependent rates (Section 5.4), while it is well-known that protein transition kinetics exhibit non-Arrhenius temperature dependence, the reference MD values are from a DESRES trajectory at a single temperature, so whether the dependency or trend is captured correctly remains unknown from this benchmark.

---

> ### Author Rebuttal · Authors · 2026-03-31
>
> We thank the reviewer for the detailed review and we hope the following responses address each point thoroughly.
>
> &nbsp;
>
> **Core contribution**
>
> We refer the reviewer to our response to Reviewer DyQi on a similar concern that "pLM embeddings will undoubtedly enhance performance”.
>
> &nbsp;
>
> **Comparison with DeepJump**
>
> While DeepJump is trained and evaluated on the same datasets as PLaTITO, our contributions differ:
> - We focus on the data efficiency enabled by incorporating pretrained representations and ITO learning.
> - We present a more rigorous evaluation spanning equilibrium thermodynamics, kinetics, cryptic pockets and temperature dependence.
> - In DeepJump, a separate model is trained for each time lag and simulation temperature is completely ignored, limiting the ability to capture temperature-dependent kinetics.
>
> &nbsp;
>
> **Why reported MFPTs are faster than MD**
>
> As the reviewer correctly mentions, PLaTITO estimates faster MFPTs than those obtained from MD. This is most likely attributable to two compounding factors:
> 1. Imperfect training of the transfer operator, which introduces a variational bias toward faster transitions.
> 2. Non-Markovian effects from coarse-graining, which are not fully captured by a Markovian transfer operator at the $C_{\alpha}$ level.
>
> Based on prior ITO work (Schreiner et al., 2023; Diez et al., 2025a), **we expect factor (1) to be the dominant source of accelerated MFPTs**. The variational principle of conformational dynamics (Nüske et al., 2014) guarantees that imperfect approximations of the transfer operator systematically underestimate relaxation timescales and our model scaling ablations (presented in response to Reviewer YFLU) show that performance improves consistently with increased capacity, suggesting that timescale estimates would similarly improve with further scaling.
>
> Regarding promising next steps: **scaling training data and model capacity should directly reduce the bias from factor (1), while exploring scalable all-atom architectures would address factor (2)**.
>
> &nbsp;
>
> **Model scaling**
>
> We refer the reviewer to our model scaling experiments presented to reviewer YFLU.
>
> &nbsp;
>
> **On generating MD data for temperature-dependent kinetics validation**
>
> We appreciate this suggestion. However, we have estimated the cost required and believe it is not feasible within the rebuttal period.
> To reliably estimate rates from MD simulations, the total simulation time must be at least an order of magnitude longer than the slowest rate characterizing the process. Using PLaTITO's estimated rates as a lower bound on the expected MD rates, the required simulation times per temperature are at least:
> - BBA: [200, 200, 100, 10, 10] $\mu s$ across five temperatures = 520 $\mu s$ in total
> - Villin [300, 100, 50, 10, 10] $\mu s$ across five temperatures = 470 $\mu s$ in total
>
> Assuming a throughput of 1.5 $\mu s$/day, these values translate to 346 and 313 GPU days respectively or approximately 16,000 GPU-hrs in total. This is way more than  the total compute time used to train all PLaTITO variants presented in the paper.
>
> &nbsp;
>
> **Temperature extrapolation for Trp-cage**
>
> As the reviewer mentions, Trp-cage was evaluated at a higher temperature because its reference MD temperature falls outside our training range (320–450 K). Outside this range, we expect smooth extrapolation since temperature is given to the model as a continuous variable, but we acknowledge that extrapolation behavior is not formally guaranteed and performance will degrade for temperatures far from the training distribution.
>
> &nbsp;
>
> **Minor points**
>
> We thank the reviewer for raising these points that can improve the quality of the manuscript.
>
> 1. The RMSD and Coverage values reported for BioEmu in Table 1 were indeed erroneously copied from the PLaTITO+Struct row.
>   | Model | RMSE (↓) | Coverage (↑) |
>   |-------|----------|--------------|
>   | BioEmu (corrected) | 1.389±0.346 | 0.594±0.175 |
>   | BioEmu (paper) | 1.213±0.348 | 0.655±0.158 |
>
>    We note that this change further strengthens our results, as PLaTITO-Big's improvement over BioEmu is larger than originally reported.
>
> 2. We thank the reviewer for catching the CHARMM22 → CHARMM22* correction.
>
> 3. Fast-folding dataset is available upon request for research purposes and we were unable to obtain the MD data for chignolin at the time of submission. We are actively working to get access and will include the results in the revised manuscript if possible. We also mention that Chignolin is the smallest system in the dataset (10 residues) and our results show no systematic dependence of performance on protein size, so its omission is unlikely to affect our conclusions.
>
> &nbsp;
>
> We hope these clarifications and our new ablations presented across all our responses (pLM scaling, model scaling, additional baselines) address the reviewer’s concerns.

---

> > ### Author Rebuttal · Reviewer_1FUJ · 2026-03-31
> >
> > I appreciate the authors’ response and their clarification regarding the source of the error. I have one follow-up question concerning the validation of temperature-dependent kinetics.
> >
> > The MD simulation times provided by the authors relate to **folding** kinetics. However, my original question referred to the substantially faster **BBA/Villin unfolding timescales**, which appear to be accessible via unbiased/biased simulations. At present, with only a single reference data point, the claim of reproducing temperature-dependent kinetics remains insufficiently supported. Could the authors provide further analysis or clarification on this point? I will adjust my score after the authors' response to this.

---

> > > ### Author Response · Authors · 2026-04-01
> > >
> > > We thank the reviewer for the follow-up and the important point raised.
> > >
> > > We agree that claiming to reproduce temperature-dependent kinetics would require point-by-point comparison between PLaTITO's predicted timescales and MD reference timescales at multiple temperatures. We want to clarify that **this is not the claim we make**. Our claims regarding temperature-dependent kinetics are:
> > >
> > > - The temperature-dependent kinetics learned by PLaTITO are non-Arrhenius, consistent with complex rugged folding free energy landscapes (Contribution 3 in the manuscript). We support this claim by plotting the predicted timescales as a function of inverse temperature where we observe a clear curvature rather than a straight line which is qualitatively consistent with experimental observations (Alexander et al., 1992; Tan et al., 1996; Scalley & Baker, 1997).
> > >
> > > - PLaTITO systematically predicts faster timescales than MD, which aligns with the variational principle of conformational dynamics (Nüske et al., 2014) and is consistent with previous ITO work (Schreiner et al., 2023; Diez et al., 2025a). This is observed in the single temperature where MD reference data is available, for both folding and unfolding.
> > >
> > > We will carefully revise the manuscript **to ensure we avoid claiming reproduction of temperature-dependent kinetics anywhere in the paper**.
> > >
> > > We also agree that the unfolding timescales of BBA and Villin would be more accessible than the folding timescales to estimate from MD. Using the same methodology as before:
> > >
> > > - BBA: [20, 20, 25, 30, 40] $\mu s$ across five temperatures = 135 $\mu s$ in total
> > > - Villin: [20, 20, 25, 30, 40] $\mu s$ across five temperatures = 135 $\mu s$ in total
> > >
> > > These values translate to around 4,300 GPU-hours. While cheaper than the folding simulations, this remains **infeasible within the rebuttal timeline**. However, we fully agree that such a dataset would be valuable for quantitatively evaluating PLaTITO's temperature-dependent predictions and **we plan to pursue this in future work**.

---

### Official Review · Reviewer_YFLU · 2026-03-18

**Soundness:** 3
**Presentation:** 3
**Significance:** 3
**Originality:** 2
**Overall Recommendation:** 4
**Confidence:** 3

**Summary:**

This paper introduces PLaTITO, which adds pre-trained protein language model (pLM) embeddings to the TITO framework for generative molecular dynamics, aiming to improve generalization to unseen proteins with less training data. The scaled-up variant, PLaTITO-Big, matches or beats BioEmu on fast-folder equilibrium benchmarks while using substantially less MD data and compute. The paper also tests structure embeddings, LLM-derived annotations, temperature-dependent kinetics, and cryptic-pocket sampling.

**Compliance With Llm Reviewing Policy:**

Affirmed.

**Final Justification:**

I support a weak accept. The paper presents a simple and practically meaningful extension of TITO by incorporating residue-level pLM embeddings, and the empirical results are promising. In particular, the claim that PLaTITO-Big matches or outperforms BioEmu while using substantially less MD data and compute gives the work clear practical significance.

My main concern in the original review was whether the benefit of pLM embeddings was sufficiently disentangled from the effect of increased model capacity. The rebuttal addressed this concern with additional ablations that vary pLM choice and model size separately, which strengthens the support for the paper’s central claim. As a result, my assessment of soundness improved from fair to good.

My view on originality is largely unchanged, as I still see the paper more as a strong extension of an existing framework than a fundamentally new conceptual contribution. Overall, I find the paper technically solid, clearly presented, and practically useful, and the rebuttal positively changed my soundness evaluation by resolving my main concern.

**Key Questions For Authors:**

See above weakness.

**Limitations:**

Yes

**Strengths And Weaknesses:**

## Strengths

1. The core methodology is simple, well motivated, and empirically effective. Conditioning TITO on residue-level pLM embeddings is a clean way to inject biologically meaningful prior information into operator learning, and the progression from TITO to PLaTITO to PLaTITO-Big suggests that this choice yields practically relevant gains.

2. Experimental result that a TITO-style model can match or outperform a strong equilibrium baseline (BioEmu) while using substantially less MD data and compute is interesting and promising.

## Weaknesses

1. Despite significant effects, the proposed method feels more like a strong and well-executed extension of an existing framework than a new conceptual paradigm. This does not make the paper weak, but its novelty is somewhat less compelling.

2. Because pLM embeddings are the central intervention of the paper, I would like to see a more systematic ablation of the representation itself. Current results show that the chosen pLM-based setup works better, but they do not yet fully establish whether the gain is robust across different pLM backbones (e.g., use ESM-2 as backbone) or whether it scales consistently with pLM size/strength. Although the paper provides an analysis on pLM scaling (300M → 6B), but it is confounded with a simultaneous increase in model capacity (3M → 19M parameters), making it hard to disentangle the two factors. Comparisons across alternative protein language models and smaller versus larger pLMs would make the paper's main claim substantially stronger.

---

> ### Author Rebuttal · Authors · 2026-03-31
>
> We thank the reviewer for the constructive feedback and we address each point separately below.
>
> &nbsp;
>
> **Novelty**
>
> - The primary contribution is the demonstration that **near-zero-cost pLM embeddings can dramatically improve the data efficiency and OOD generalization** of coarse-grained protein molecular dynamics. Indeed, the ITO learning framework (Schreiner et al., NeurIPS 2023) and its transferable variant (Diez et al, 2025) is a foundation, and we do not claim any new architectural advances in this paper. However, the claim that PLaTITO-Big outperforms BioEmu while using nearly an order of magnitude less MD data and compute is a practically significant finding for the community, since the value of surrogates rely on their upfront cost (training compute/data), inference cost and accuracy. This work makes significant advances on all three parameters compared to the current state of the art (BioEmu, Lewis et al, Science, 2025).
>
> - To our knowledge, **no prior work has demonstrated that pLM embeddings improve protein ensemble generation or any other related task in GenMD**. Here, we establish empirically how much these embeddings help in a severely data-limited regime and whether they enable competitive generalization without system-specific fine-tuning.
>
> - We also demonstrate that **ITO learning is more data-efficient than Boltzmann Emulators**, even when the target is the equilibrium distribution. We implemented Emu, a model architecturally identical to TITO but trained to sample directly from $p(x | S, T)$:
>
>   | Model | MAE (↓) | RMSE (↓) | Coverage (↑) |
>   |-------|---------|----------|--------------|
>   | Emu | 1.305±0.378 | 1.639±0.406 | 0.529±0.112 |
>   | TITO | **1.068**±0.272 | **1.382**±0.302 | **0.590**±0.111 |
>
>   TITO outperforms Emu across all metrics despite using exactly the same architecture, data and compute budget indicating that exploiting the auto-correlation structure of MD data lowers the data and compute requirements of training MD surrogates.
>
> &nbsp;
>
> **Disentangling pLM size/strength from model capacity**
>
> We have run additional experiments that disentangle the two factors by changing pLM size/strength and model capacity independently.
>
>  *Effect of pLM choice at fixed model capacity (3M parameters)*
>
>   | Family | Size | Dim | MAE (↓) | RMSE (↓) | Coverage (↑) |
>   |--------|------|-----|---------|----------|--------------|
>   | - | - | - | 1.068±0.272 | 1.382±0.302 | 0.590±0.111 |
>   | ESMC | 300M | 960 | **0.949**±0.269 | **1.228**±0.328 | **0.651**±0.151 |
>   | ESMC | 600M | 1152 | 0.964±0.268 | 1.307±0.303 | 0.578±0.072 |
>   | ESMC | 6B | 2560 | 0.976±0.251 | 1.261±0.268 | 0.611±0.080 |
>   | ESM2 | 650M | 1280 | 0.968±0.261 | 1.256±0.290 | 0.579±0.090 |
>
> &nbsp;
>
>  *Effect of pLM choice at fixed model capacity (19M parameters)*
>
>   | Family | Size | Dim | MAE (↓) | RMSE (↓) | Coverage (↑) |
>   |--------|------|-----|---------|----------|--------------|
>   | ESMC | 300M | 960 | 0.840±0.228 | 1.124±0.262 | **0.674**±0.105 |
>   | ESMC | 6B | 2560 | **0.824**±0.170 | **1.099**±0.212 | 0.666±0.136 |
>
> &nbsp;
>
> - In all cases, regardless of the model family and size, **incorporating pLM embeddings during training improves generalization performance**, further strengthening our claim that pretrained pLM embeddings encode information useful for dynamics.
> - Interestingly, the benefit of larger pLMs depends on model capacity. In PLaTITO-3M, the smaller ESMC-300M performs best, suggesting that PLaTITO-3M model lacks sufficient capacity to fully exploit the higher-dimensional embeddings of ESMC-6b. When scaling to PLaTITO-19M, we see that the larger embeddings from ESMC-6b yield better MAE and RMSE. This indicates that **scaling pLM representations and model capacity are complementary** since higher-dimensional embeddings require increased model capacity to be fully leveraged.
>
> &nbsp;
>
> *Effect of model scaling when keeping a fixed pLM (ESMC-6B)*
>
>  | Parameters | MAE (↓) | RMSE (↓) | Coverage (↑) |
>  |------------|---------|----------|--------------|
>  | 1M | 1.253±0.366 | 1.605±0.459 | 0.569±0.101 |
>  | 3M | 0.976±0.251 | 1.261±0.268 | 0.611±0.080 |
>  | 19M | 0.824±0.170 | 1.099±0.212 | 0.666±0.136 |
>  | 36M | **0.811**±0.213 | **1.066**±0.252 | 0.664±0.111 |
>  | 100M | 0.858±0.266 | 1.115±0.280 | **0.668**±0.115 |
>
> &nbsp;
>
> - We observe that performance improves consistently from 1M to 36M parameters, with a slight degradation at 100M, **consistent with scaling laws** where further capacity scaling requires a corresponding increase in training data.
> - Notably, PLaTITO-36M slightly outperforms the PLaTITO-19M (PLaTITO-Big) configuration reported in the paper, suggesting that **there is room for further improvement by jointly scaling model capacity and training data**.
>
> In total, these new ablations further confirm that both factors (pLM embeddings and model capacity) contribute independently to performance gains.
>
> &nbsp;
>
> We hope these additional experiments will address the concerns of the reviewer.

---

> > ### Author Rebuttal · Reviewer_YFLU · 2026-04-03
> >
> > Thank you for the thoughtful rebuttal and the additional experiments, and I appreciate the authors’ effort in responding carefully. The pLM size/strength disentangling results address my concern. I will increase the soundness score and maintain my initial overall recommendation score.

---

> > > ### Author Response · Authors · 2026-04-03
> > >
> > > We thank the reviewer for the positive feedback and for increasing the soundness score.
> > >
> > > We also want to add that during the rebuttal period we evaluated two additional competitive baselines (ConfDiff and Str2Str) on equilibrium sampling and **PLaTITO still achieves the best performance, further demonstrating its importance for the community** (details in our response to Reviewer xcH3).
> > >
> > >
> > > Given that both concerns raised in the original review have been addressed along with these new results, we kindly ask the reviewer to consider whether an increase to the overall recommendation score might also be possible.

---

### Decision · Program_Chairs · 2026-04-30

**Decision:**

Accept (regular)

**Comment:**

This paper presents PlaTITO, an extension of the TITO framework with a protein language model embedding for better generalization across proteins. PlaTITO uses an ESM embedding which is shown to improve generalization. The method is coarse-grained and trained on the mdCATH dataset.

The paper received three high quality reviews and one low quality review with final overall scores of 5 4 4 and 3. Three of the four reviewers engaged constructively during the discussion period with two explicitly raising their scores after the rebuttal. The fourth reveiwer DyQi (score 3) did not respond after the initial review. I read through this review, determined that the majority of the points were adequately addressed by the authors.

All the reviewers agree that the paper is well-written and clearly presented. The design choice is well motivated and sensible, with a strong evaluation and experimental design. While not that exciting of an idea, as primarily an extension of Tito with ESM embeddings, this is a well executed work overall.

As discussed privately with the authors, it was agreed that the claim that "no prior work has demonstrated that pLM embeddings improve protein ensemble generation" is stated too broadly. I encourage the authors to revise the claim as agreed. Nevertheless, given the paper's execution and potential usefulness to the community I recommend acceptance at this time.